# An intergenerational androgenic mechanism of female intrasexual competition in the cooperatively breeding meerkat

Christine M. Drea [1,2,3 ✉], Charli S. Davies [1,3,6], Lydia K. Greene [1,3,7], Jessica Mitchell[1,3,8], Dimitri V. Blondel[1,3,9], Caroline L. Shearer [1], Joseph T. Feldblum [1,10], Kristin A. Dimac-Stohl[1,2], Kendra N. Smyth-Kabay[1,3,11] & Tim H. Clutton-Brock [3,4,5]

Female intrasexual competition can be intense in cooperatively breeding species, with some dominant breeders (matriarchs) limiting reproduction in subordinates via aggression, eviction or infanticide. In males, such tendencies bidirectionally link to testosterone, but in females, there has been little systematic investigation of androgen-mediated behaviour within and across generations. In 22 clans of wild meerkats (*Suricata suricatta*), we show that matriarchs 1) express peak androgen concentrations during late gestation, 2) when displaying peak feeding competition, dominance behaviour, and evictions, and 3) relative to subordinates, produce offspring that are more aggressive in early development. Late-gestation antiandrogen treatment of matriarchs 4) specifically reduces dominance behaviour, is associated with infrequent evictions, decreases social centrality within the clan, 5) increases aggression in cohabiting subordinate dams, and 6) reduces offspring aggression. These effects implicate androgen-mediated aggression in the operation of female sexual selection, and intergenerational transmission of masculinised phenotypes in the evolution of meerkat cooperative breeding.

[1] Department of Evolutionary Anthropology, Duke University, Durham, NC 27708, USA. [2] Department of Biology, Duke University, Durham, NC 27708, USA. [3] Kalahari Research Trust, Kuruman River Reserve, Northern Cape, South Africa. [4] Department of Zoology, University of Cambridge, Cambridge CB2 3EJ, UK. [5] Mammal Research Institute, University of Pretoria, 0002 Pretoria, South Africa. [6] School of Biological Sciences, University of East Anglia, Norwich Research Park, Norwich NR4 7TJ, UK. [7] Duke Lemur Center, Duke University, Durham, NC 27705, USA. [8] Nuffield Centre for International Health and Development, University of Leeds, Leeds, West Yorkshire LS2 9JT, UK. [9] Department of Biology, North Carolina Wesleyan College, Rocky Mount, NC 27804, USA. [10] Department of Anthropology and Society of Fellows, University of Michigan, Ann Arbor, MI 48109, USA. [11] Boston Consulting Group, Bethesda, MD 20814, USA. ✉email: cdrea@duke.edu

In cooperatively breeding species, nonbreeding helpers assist in raising young born to dominant breeders, such that both sexes experience pronounced reproductive skew. Whereas the functional explanations for why helpers forfeit reproduction to assist others implicate benefits gained through kin selection or inclusive fitness (augmented through mutualism or reciprocity)[1,2], the proximate physiological and behavioural mechanisms of reproductive monopoly or suppression can be elusive[3,4], particularly if supporting evidence is strictly correlational. Studies on the neuroendocrine mediators of cooperative breeding are often focused on reproductive suppression, targeting the role of oxytocin, vasopressin, prolactin, and testosterone in alloparental care[5,6] or the role of glucocorticoids, progestins and oestrogens in subordinate infertility[7,8]. Less is known about the neuroendocrine mediators of reproductive monopoly.

Whereas androgen-mediated power struggles over reproductive opportunities, including infanticide, are commonplace among male mammals[9–11], the same is uncharacteristic of their female counterparts[12]. Nevertheless, in select species, such as spotted hyaenas (*Crocuta crocuta*)[13], meerkats (*Suricata suricatta*)[14,15] and other communal or cooperative breeders[16–18], females express unusually high androgen concentrations associated with female dominance or breeding status: female values for androstenedione and/or testosterone approach, meet or exceed those of conspecific males, either year-round or in association with reproduction. Although typically adversative to females[19], androgens in these species may be key to explaining aggressively mediated, female intrasexual competition, leading to differential reproductive advantage. We examine this possibility in the meerkat.

The meerkat is a social mongoose and obligate cooperative breeder that displays a despotic (rather than linear) hierarchy, with long-term stability. A single matriarch holds the primary power position and leadership role and, along with a dominant male partner, largely monopolises aseasonal, asynchronous reproduction within a clan of up to 50 subordinate individuals[14]. Matriarchs can produce 3–7 pups per litter, up to four times per year, and can breed for up to 10 years[14]. Offspring of both sexes reach sexual maturity at ~1 year and typically disperse at ~2 years. Unlike many other cooperative breeders, all adult meerkats are physiologically capable of breeding, but adhere to a limited control model of reproductive skew. Because competitiveness underlies reproductive success[14,20], the degree of skew varies socially and environmentally (Supplementary Fig. 1)[21,22]. Any dam can commit infanticide, but evictions of subordinate females are typically performed by the matriarch. Status-related differences in concentrations of glucocorticoids, progestins and oestrogens[20,22–24] cannot explain the subordinates' apparent reproductive suppression[22]. Instead, the female meerkat's aggressive nature[14,25], coupled with exceptionally high androgen concentrations, particularly whilst pregnant[15], raise the possibility of an androgenic mechanism of female reproductive monopoly. We propose that female meerkats are naturally physiologically and behaviourally masculinised (sensu Phoenix and colleagues[26]) by the actions of their own androgens.

Although multiple genetic and non-genetic factors contribute to mammalian sexual differentiation[27], two modes of hormone action likely underlie this phenomenon[26]. (i) Concurrent and reversible activational effects of late to post-gestational androgens could facilitate the matriarch's aggression at an opportune time, both for expelling reproductive competitors (when access to resources is critical) and for protecting vulnerable offspring (when threat of infanticide is high[28]). (ii) Concomitantly, reflecting a temporal prenatal sequence in endocrine action (starting with gonadal differentiation, then internal and external genital differentiation, ending with brain differentiation[29]),

maternal androgens[30–32] could have permanent and delayed organisational effects on developing offspring. Specifically affecting the late-developing neural circuitry of foetal daughters could promote differences in their future aggressiveness, based on maternal status, without interfering with reproductive function[30,33,34]. Thus, as may be the case in various species[31,32,35,36], meerkat daughters might be differentially advantaged to later compete for dominant status and, ultimately, gain reproductive success. Here, we combine normative and experimental data to ascertain female meerkat phenotypes and test for both activational and organisational effects on behaviour that could reveal a heritable/epigenetic, androgenic mechanism of female reproductive competition.

The Female Masculinisation Hypothesis leads to three initial predictions about normative, status-related and temporal patterns in meerkat behavioural endocrinology. Whereas androgens might rise slightly in early gestation in many mammals, those species in which females are unconventionally aggressive also show a signature late-term increase[13,37]. Thus, in addition to 1a) greater androgen concentrations in dominant than subordinate meerkat dams[14,15], we expect 1b) androstenedione and/or testosterone concentrations to increase across gestation and potentially into the early postpartum period, 2a) a corresponding rise in the matriarchs' competitive or androgen-mediated behaviour (directed at any clan member) and 2b) an increase in the prosocial or submissive behaviour she receives from subordinates. We also expect 3) an organisational effect of differential, late-term androgen exposure on offspring aggression, potentially evident in both sexes, well before the onset of activational effects at puberty[32,38], namely in pups (1–3 months) and/or juveniles (3–6 months).

An experimental test of this hypothesis requires manipulating the endocrine milieu, in our case by interfering with androgen action via receptor blockade, specifically in dominant dams late in gestation and into the early postpartum period. Such a manipulation gives rise to an additional three sets of predictions. Treated matriarchs should concurrently 4a) diminish their competitive and dominance-related behaviour, specifically, 4b) experience reduced social centrality within the clan, indicative of altered social dynamics, and 5) increase competition between or diminish submission in cohabiting subordinate dams. They should also ultimately 6) bear offspring that show reduced prepubertal aggression relative to the offspring of control matriarchs.

Here, we present data that meet these predictions. We provide normative and experimental, intergenerational evidence for the functional significance of androgens in female-female competition. Because female dominance in meerkats underlies dramatic female reproductive skew, we suggest androgen-mediated aggression as a mechanism for female sexual selection, potentially including heritable, maternal endocrine effects affecting female reproductive trajectories in the evolution of cooperative breeding.

## Results

**Gestational androgens reflect status and increase over pregnancy.** We studied well-habituated, wild meerkats in the Kalahari Desert to first examine normative behavioural endocrine patterns (see "Methods" section and Tables 1–3). Consistent with prediction 1a, androgen concentrations in dominant females during and post pregnancy exceeded those in subordinate females (generalised linear mixed model (GLMM): androstenedione $\chi^2_{1,31} = 50.49$, $P < 0.001$; testosterone $\chi^2_{1,32} = 6.39$, $P = 0.011$; linear mixed model (LMM): faecal androgen metabolites (fAm) $\chi^2_{1,7} = 7.60$, $P = 0.006$; Fig. 1a–c and Supplementary Table 1). Consistent with prediction 1b, overall female androgen

**Table 1 Metadata on serum and faecal samples from meerkat dams.**

| Sample type | Dam category | Dam features | | | n samples by stage | | | | n samples (total) |
|---|---|---|---|---|---|---|---|---|---|
| | | n | Mean age (year) | n litters | EP[a] | MP | LP | PP[b] | |
| Serum | Dominant control | 18 | 4.3 | 27 | 6 | 6 | 15 | [1] | 27 |
| | Subordinate control | 17 | 2.2 | 22 | 5 | 8 | 9 | [3] | 22 |
| | Total unique used | 35 | 3.3 | 49 | 11 | 14 | 24 | [4] | 49 |
| Faecal | Dominant control[c] | 20 | 4.9 | 31 | (4) | 23 | 39 | 26 | 88 |
| | Subordinate control | 16 | 1.5 | 18 | (5) | 8 | 23 | 19 | 50 |
| | Dominant treated[c] | 6 | 5.4 | 7 | – | – | 17 | 1 | 18 |
| | Total unique | 36 | 3.8 | 51 | (9) | 31 | 79 | 46 | 156 |

[a]Too few faecal samples (in parentheses) were obtained for analysis in early pregnancy (EP), relative to mid pregnancy (MP), late pregnancy (LP), and postpartum (PP), and are thus excluded from the total sample numbers reported (last column).
[b]Females typically were not captured during early PP, providing too few serum samples [in brackets] for analysis, and are thus excluded from the total sample numbers reported.
[c]All endocrine data obtained pre-implantation from matriarchs that would be later treated (i.e., dominant treated) contributed to the data from dominant control animals.

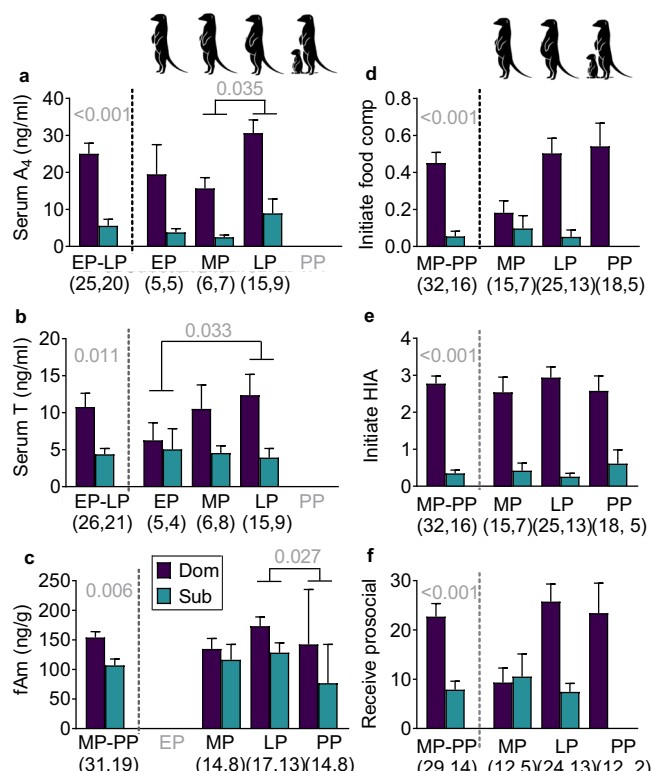

**Fig. 1 Normative status differences in androgen concentrations and behavioural rates from early pregnancy to post-parturition in wild meerkats.** Concentrations of **a** serum androstenedione ($A_4$), **b** serum testosterone (T) and **c** faecal androgen metabolites (fAm), as well as hourly rates of **d** initiating food competition, **e** initiating high-intensity aggression (HIA) and **f** receiving prosocial overtures vary in meerkat dams by status (dominant: purple; subordinate: teal) and by pregnancy stage (early, mid and late pregnancy, plus the first two weeks postpartum: EP, MP, LP and PP, respectively). Vertical dashed lines separate mean + s.e. values overall from those by pregnancy stage. Analyses performed on 187 biological samples from 53 dams and 1174 focal observations during 48 pregnancies of 37 dams. GLMMs (or LMM in **c**), with two-tailed $\chi^2$, showed greater endocrine and behavioural values in dominant than subordinate dams throughout; least significant differences (LSD) showed greatest endocrine values in LP (for dominant and subordinate dams combined); significant P values in grey. (Icons of pregnancy stages drawn by S. Bornbusch). See Supplementary Fig. 2 for the distribution of raw endocrine data. Source data are provided as a Source Data file.

concentrations changed significantly from early, mid, to late pregnancy (EP, MP and LP, respectively) and to early postpartum (PP), defined as the first two weeks after birth (GLMM: androstenedione $\chi^2_{2,12} = 9.50$, $P = 0.008$; testosterone $\chi^2_{2,11} = 9.50$, $P = 0.009$; LMM: fAm $\chi^2_{2,7} = 7.19$, $P = 0.027$), reaching their peak during LP (least significant difference (LSD): androstenedione LP > MP, $t = -2.86$, $P = 0.035$; testosterone LP > EP, $t = -2.94$, $P = 0.033$; fAm LP > PP, $t = 2.61$, $P = 0.027$; Fig. 1a–c and Supplementary Fig. 2 and Supplementary Table 1). There were no significant interactions between status and pregnancy stage, indicating that dominant females maintained their androgen advantage over subordinates consistently across gestation.

**Gestational behaviour reflects status and changes over pregnancy.** As resources are limited in the Kalahari, food competition —a subset of intense aggressive behaviour—and evictions are particularly salient to female reproductive competition. Consistent with prediction 2a, dominant dams more frequently initiated food competition (GLMM: $\chi^2_1 = 12.51$, $P < 0.001$; Fig. 1d), initiated more intense overall aggression ($\chi^2_1 = 35.39$, $P < 0.001$; Fig. 1e) and engaged in more status-advertising scent marking ($\chi^2_1 = 5.31$, $P = 0.021$) than did subordinate dams across pregnancy (Supplementary Table 2). Consistent with prediction 2b, dominant dams also received more prosocial overtures from clan members than did subordinate dams ($\chi^2_1 = 14.72$, $P < 0.001$; Fig. 1f). When only matriarchs were included in the time-course analysis, rates of initiating food competition ($\chi^2_2 = 9.78$, $P = 0.008$) and receiving prosocial overtures ($\chi^2_2 = 18.80$, $P < 0.001$; LSD: LP > MP $t = -3.14$, $P = 0.004$) increased significantly across pregnancy. The matriarch's androgen concentrations (Fig. 1a–c) and dominance interactions (Fig. 1d–f) thus peaked in LP, when the evictions of subordinate females, inferred from day-to-day changes in clan composition, also occur most frequently[28].

**Maternal status differentiates normative infant aggression.** Consistent with prediction 3, analysis of behavioural ontogeny revealed powerful maternal effects and heretofore unknown patterns in early meerkat aggression, suggestive of organisational effects of androgens. Exceptionally, the offspring from both sets of mothers showed no sex difference (i.e., females were just as aggressive as males) and those from dominant mothers initiated more aggression than those from subordinate mothers throughout their first six months (GLMM: $\chi^2_2 = 13.71$, $P = 0.001$; Fig. 2 and Supplementary Fig. 3a, b and Supplementary Table 3). Moreover, pup aggression was initially pronounced, coincident with peak

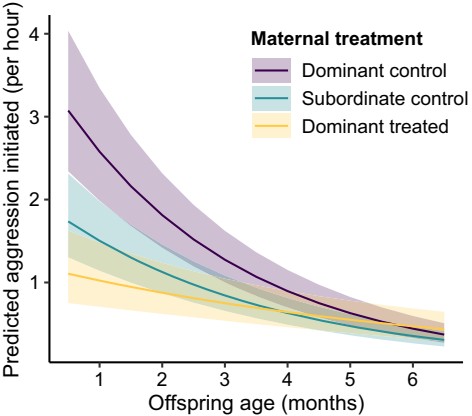

**Fig. 2 Different model-predicted rates of offspring aggression by age, based on maternal status and treatment condition.** The initiation of aggression by meerkat offspring is significantly predicted by their age during the first 6 months of life ($P < 0.001$) and by their mother's status and treatment condition during pregnancy ($P = 0.001$): dominant control (purple), subordinate control (teal), and dominant treated (orange). Solid lines represent model-predicted rates (frequency/hour), with 95% confidence intervals. Analyses used GLMMs, with two-tailed $\chi^2$, based on 103 offspring, observed during 5661 focal sessions. See Supplementary Fig. 3 for the distribution of raw data.

competition for attention from and feeding by helpers, then declined with advancing age to juvenility (GLMM: $\chi^2_1 = 264.70$, $P < 0.001$), alongside the acquisition of independent feeding. Reflecting the different initial rates of aggression, there was a steeper negative slope in the offspring from dominant than subordinate mothers (GLMM: $\chi^2_2 = 21.64$, $P < 0.001$; LSD: $t = 2.89$, $P = 0.011$; Fig. 2).

**Flutamide alters key behaviour in matriarchs.** We next examined activational effects of an antiandrogen on these same behavioural and endocrine patterns. Flutamide is a model, mammalian androgen receptor blocker that produces comparable inhibitory effects on androgen-mediated traits in both sexes, including during pregnancy[39–42]. To experimentally test the Female Masculinisation Hypothesis, we used procedures previously validated in male meerkats[43] to administer this antiandrogen to a subset of dominant dams late in gestation and into the early postpartum period (see "Methods" section and Supplementary Fig. 4).

Consistent with prediction 4a, treated matriarchs initiated less food competition than did control matriarchs (GLMM: $\chi^2_1 = 7.39$, $P = 0.007$; Fig. 3a and Supplementary Table 4)—a finding that reprised treatment effects in less aggressive, adult male conspecifics[43] (Supplementary Fig. 5). Likewise, treated matriarchs less frequently scent marked ($\chi^2_1 = 6.94$, $P = 0.008$; Fig. 3b) and received fewer prosocial overtures ($\chi^2_1 = 5.73$, $P = 0.017$; Fig. 3c) or submissive acts ($\chi^2_1 = 4.26$, $P = 0.039$; Fig. 3d) from clan members than did control matriarchs. Whereas, during LP, control females evicted 33% of subordinate females, treated females evicted only 13% (nearly 60% fewer; Fig. 3e), but so few evictions in the experimental group precluded statistical modelling.

Consistent with prediction 4b, in a subset of four clans in which each matriarch experienced, in random order, both a control and a treated pregnancy (such that respective clan sizes, territories, memberships and social dynamics were matched), antiandrogens significantly reduced the frequency of associations between the matriarch and adult clan members (LMM, $t_{51.15} = -3.04$, $P = 0.004$; Fig. 3f and Supplementary Table 5).

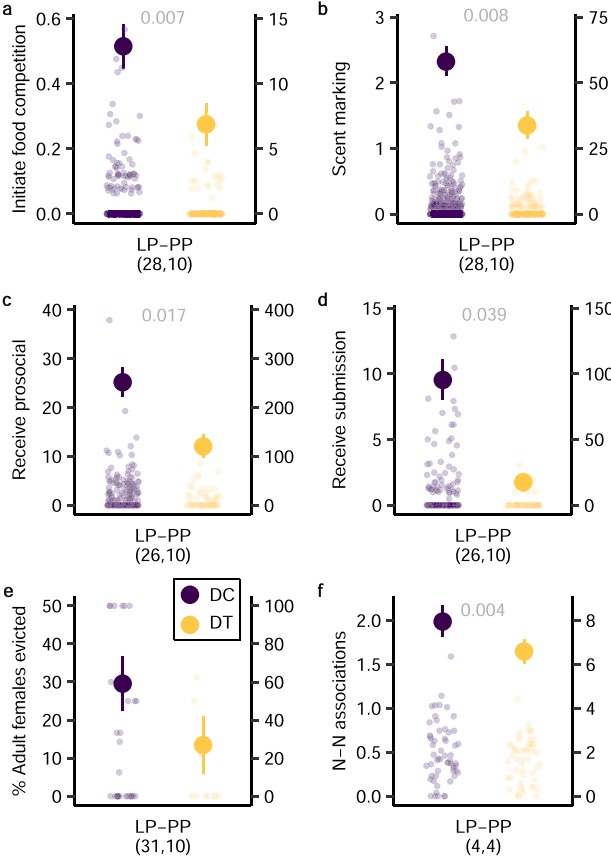

**Fig. 3 Effects of antiandrogen treatment on the behaviour of dominant dams.** Antiandrogen treatment of matriarchs during late-pregnancy (LP) and early postpartum (PP) reduces hourly rates of **a** initiating food competition, **b** scent marking, **c** receiving prosocial overtures and **d** receiving submissive acts relative to control matriarchs. Dominant control (DC, purple); dominant treated (DT, orange). Mean ± s.e. on the left y axis; raw data on the right y axis (to accommodate zero inflation). Analyses used GLMMs, with two-tailed $\chi^2$, based on 32 pregnancies of 22 dominant dams, observed during 1044 focal sessions. **e** The percentage of adult females evicted also appeared to be reduced in DT relative to DC dams, but analysis was precluded. Lastly, treatment reduced **f** hourly rates (scans/partner/hr) of dyadic nearest-neighbour (N-N) associations. For the LMM analysis, matriarchs were matched across control and treated conditions in four clans, with sample sizes ($n = 110$) reflecting all possible adult dyads involving each matriarch, and a null expectation of no relationship between predictor terms and rate of proximity ($P$ values represent two-tailed tests). Accordingly, relative to DC dams, DT dams were in close proximity to another adult clan member during ~1 fewer scan/individual/2 h observation (in a clan with a mean of 14 adults, this value corresponds to a predicted difference of ~8 N-N associations/h). Values in parentheses represent the numbers of **a**-**e** pregnancies or **f** matched clans sampled; significant $P$ values in grey. Source data are provided as a Source Data file.

Thus, whether directly reducing aggression and scent signalling or indirectly altering chemosignal composition, antiandrogens impacted how matriarchs behaved toward and were perceived by clan members. Within a mere three-week window, treated dams behaved in a more docile manner than is characteristic of matriarchs, commanded less deference from clan members, appeared to evict fewer females and became less socially central within their clan.

By contrast, treated matriarchs showed no difference relative to controls in non-target or neutral behaviour, such as vigilance, a sexually differentiated (male > female), but seemingly androgen-

independent, behaviour[43,44] (Supplementary Fig. 6a, b) and digging for food, a sexually undifferentiated behaviour (Supplementary Fig. 6c, d). Differences in vigilance frequency (control > treated: GLMM: $\chi^2_1 = 4.25$, $P = 0.039$) were negated by reverse differences in vigilance duration (control < treated; LMM: $\chi^2_1 = 5.23$, $P = 0.022$). Moreover, neither frequency nor duration of digging varied by treatment (frequency full model GLMM: $\chi^2_1 = 1.92$, $P = 0.166$; duration full model LMM: $\chi^2_1 = 0.23$, $P = 0.629$), further dispelling any potential concern about suppressive effects of treatment. Our observed effects reflect the selective action of flutamide and are specific to behaviour routinely found to be androgen mediated across species and sexes[33,34,39,41,43].

Flutamide can also variably influence circulating concentrations of androgens and other hormones, but as in male meerkats and other species[43], it had no appreciable effect on concentrations of fAm in female meerkats across the full, three-week treatment period (LMM: $\chi^2_1 = 2.10$, $P = 0.147$; Supplementary Fig. 7a and Supplementary Table 6). Likewise, treatment had no effect on concentrations of faecal glucocorticoid metabolites (fGCm; $\chi^2_1 = 0.01$, $P = 0.942$; Supplementary Fig. 7b and Supplementary Table 6) or on the dams' weight at the end of the treatment period (linear model (LM): $t_{2,25} = 0.12$, $P = 0.906$; Supplementary Fig. 7c and Supplementary Table 6). These latter findings are consistent with the absence of any negative stress or health side effects of treatment (see "Methods" section).

**Flutamide in matriarchs alters behaviour in subordinate dams.** Concurrently pregnant subordinates represent a threat to the pregnant matriarch's reproductive interests, as competing pups can dilute the benefits provided by helpers. Because pregnancy increases the risk of eviction, subordinate dams typically maintain a low profile within the clan. It is notable, however, that dominance overthrows are often preceded by increased competition between subordinate contenders. Consistent with prediction 5, when the pregnant matriarch received antiandrogens, concurrently pregnant subordinates within her clan initiated significantly more food competition (GLMM: $\chi^2_1 = 6.48$, $P = 0.011$; Fig. 4a) and intense aggression ($\chi^2_1 = 6.55$, $P = 0.010$: Fig. 4b) whilst foraging than did subordinate dams whose pregnant matriarch was left untreated (see Supplementary Table 7 for model details). The subordinate dams' initiation of prosocial and submissive behaviour at the den also appeared to decrease (Fig. 4c, d), but the short periods spent above ground, at the den (versus whilst foraging), produced smaller sample sizes that prevented model convergence. Overall, pregnant subordinates responded to the matriarch's treatment in a manner that could potentially improve their access to resources, the defence of their own offspring and their threat to the matriarch's offspring.

**Flutamide in matriarchs alters their offspring's aggression.** Evidence of maternal effects of treatment on pups represents the ultimate test of organisational effects. Consistent with prediction 6, the offspring of antiandrogen-treated matriarchs, regardless of sex, initiated significantly less aggression than did those of control matriarchs throughout the first six months of life (GLMM: $\chi^2_2 = 13.71$, $P = 0.001$; Fig. 2 and Supplementary Fig. 3a, c and Supplementary Table 3). The age-related decline in aggression by offspring from control matriarchs was greatest relative to that by offspring from treated matriarchs (GLMM: $\chi^2_2 = 21.64$, $P < 0.001$; LSD: $t = 2.46$, $P = 0.036$). These findings confirm that maternal androgens normally reach foetal meerkats in LP and PP, presumably via the placenta and maternal milk, respectively, and influence later expression of the offsprings' behavioural phenotype. Indeed, whilst experimental elimination of the natural, status-related difference in intergenerational patterns (i.e., similar

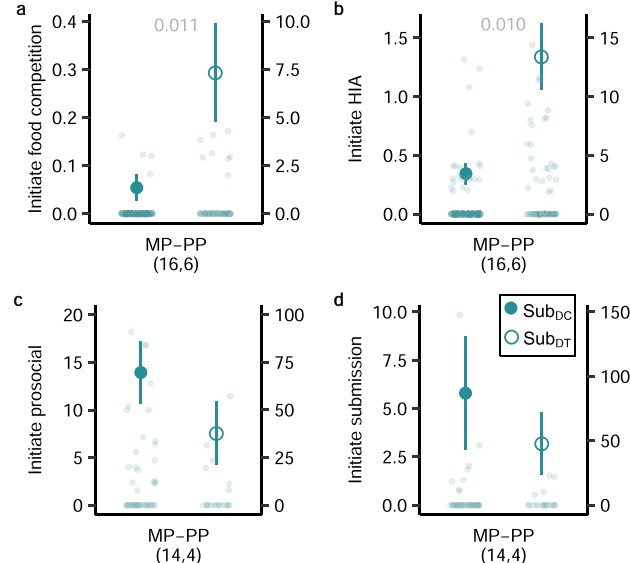

**Fig. 4 Effects of the antiandrogen treatment of dominant dams on the behaviour of concurrently pregnant subordinates within the same clan.** Subordinate dams in mid pregnancy (MP) to early postpartum (PP), that were co-dwelling with a dominant treated dam (Sub_DT) initiated higher hourly rates of **a** food competition and **b** high-intensity aggression (HIA) whilst foraging than did subordinate dams co-dwelling with a dominant control dam (Sub_DC). Sub_DC (filled teal); Sub_DT (open teal). Mean ± s.e. hourly rates on the left y axis; raw data distribution on the right y axis (to accommodate zero inflation). Analyses used GLMMs, with two-tailed $\chi^2$, based on 22 pregnancies of 22 subordinate dams, observed during 388 focal sessions. Sub_DT dams appeared to initiate fewer **c** prosocial or **d** submissive interactions whilst at the den than did Sub_DC dams, but smaller sample sizes precluded analyses. The subordinates' broader range of stages (relative to those of dominant females in Fig. 3) accommodates variability in the synchrony of concurrent pregnancies in cohabiting females. Values in parentheses represent the numbers of pregnancies sampled across locales; significant P values in grey. Source data are provided as a Source Data file.

aggression rates between offspring from treated matriarchs and subordinate controls; LSD: $t = 0.23$, $P = 0.971$; Supplementary Fig. 3b, c) also confirms the role of androgens in normative behavioural patterns, it further indicates physiological relevance of the antiandrogen dosage and treatment duration in effectuating organisational effects.

**Discussion**
This intergenerational study presents multiple elements, both from normative and experimental data sets on multiple social groups in nature, to support a unified hypothesis about female masculinisation in the cooperatively breeding meerkat. Specifically, androgenic action in female meerkats, inferred via prohibiting the cellular actions of androgens, explains both the status-related differences in adult aggression and maternal effects on offspring aggression. Whereas the potential role of aromatisation of androgens to oestrogens could not be discounted in interpreting the effects of flutamide in male meerkats[43] (given lack of knowledge about male oestrogen concentrations and their behavioural correlates), the role of aromatisation can be discounted here, given the existing information on female meerkat oestrogen concentrations[15,20,22,23]. Although unconventionally focused on wild females, our androgen-mediated behavioural findings adhere to traditional theory on sexual differentiation[29], whilst also recognising the added potential for genetic and epigenetic factors[27].

The supporting elements from the mother's generation (i.e., activational effects) include the normative physiological and behavioural differences between dominant and subordinate dams, the direct mirroring in flutamide-treated matriarchs of the normative subordinate behavioural phenotype, and the indirect effects of the matriarch's treatment on the behaviour of pregnant subordinate clan members. Both the status-related androgen advantage and temporal gestational sequence in androgen concentrations are consistent with patterns observed in other naturally masculinised females[13,17,18,36,37]. Likewise, the immediacy, specificity and potency of adult behavioural effects (e.g., reduced aggression) from androgen receptor blockade are consistent with those observed in treated male conspecifics[43] and treated females of other species[39,40], and are reversed from those (e.g., increased aggression) observed in androgen-supplemented females of other species[41,45]. Based on the behaviour of subordinate dams, we suggest that over a longer time-course than 3 weeks, antiandrogen treatment could potentially contribute to an overthrow of female dominance relations, breeding status and even reproductive outcomes.

The supporting elements from the offspring's generation (i.e., organisational effects) include the potential heritability of maternal behavioural phenotypes and the mirroring of subordinate behaviour by the descendants of treated matriarchs, evident prepubertally through 6 months of age. The normative intergenerational findings are consistent with female behavioural masculinisation in all lineages, but with dams of differing status exerting different levels of prenatal endocrine influence over their respective pups, as occurs in spotted hyaenas[36]. Moreover, the experimental intergenerational findings (i.e., reduced expression of androgen-mediated traits in both sexes of offspring) are consistent with treatment effects in other species[41,42,46].

Unlike the typical, mammalian developmental pattern of early prosociality, followed by increasing male-biased aggression, our findings on the absence of offspring sex differences and early emergence of aggression are also consistent with patterns observed in other behaviourally masculinised mammals[32,36,38]. Indeed, based on correlational data, spotted hyaenas show the same developmental[38] and intergenerational[36] patterns as meerkats; nevertheless, because maternal interventions ensure the offspring's status acquisition in the hyaena's matrilineal society[47], hyaena mothers help shape their cubs' aggressive interactions. By contrast, the cooperative rearing of meerkat pups minimises the influence of certain maternal factors[48], limiting the potential for a similar socialisation confound. The meerkat system thus better isolates the physiological influence over early aggression, illustrating the different, lineage-based potentials for androgen-mediated, female aggression—potentials that are organised in utero and activated in adulthood.

The late-gestation timing of peak androgen exposure is relevant both to activational effects on mothers that enable protection of their reproductive investment, and to organisational effects on their developing daughters, when foetal brain development is sensitive to sexual differentiation of the embryonic neural structures underlying postnatal behaviour. Timing in relation to the endocrine milieu matters because, contrary to certain unconventional species[49–53] and despite clear physiological masculinisation[15] (Fig. 1a–c), the reproductive organs of female meerkats are not masculinised. Although various endocrine routes or nonendocrine mechanisms can account for genital masculinisation of foetal females[29,31,41,42,50,53], differential genital versus behavioural masculinisation[33,51] traditionally owes to the timing, quantity and duration of foetal androgen exposure[29,33,34] and, potentially, to the type of androgen (i.e., androstenedione versus testosterone[30–32]). With regard to timing, developing daughters can be protected from extreme reproductive consequences of early- to mid-gestational androgens, for example[19], via selective enzymatic

action in utero (e.g., aromatase enhancing early placental conversion of maternal androgens to oestrogens[54]) or via as yet uncharacterised mechanisms[27]. Evidently protected from potential costs of early masculinisation, meerkats, like other vertebrate females[55,56], nonetheless bear costs of natural androgens, including increased parasitism[57] and reduced immunocompetence[58], again potentially differentiating androstenedione-mediation from testosterone-mediation. Although androgen-associated health costs may carry reproductive trade-offs during environmental stressors[22], the socio-reproductive advantages appear to far outweigh these costs.

Across mammals, we specifically lack understanding of the mechanisms and functional significance of pronounced female aggression and social dominance, both at an individual and species level. Our study addresses this gap and suggests a broader role for androgens in female mammals than previously recognised. Whereas activational effects of androgens are temporary, organisational effects typically endure, even if expression may become latent (such as when meerkat offspring settle into uniformly submissive roles as juveniles). Thus, beyond accounting for an uncharacteristically aggressive female phenotype, androgen concentrations differentiate dominant and subordinate females, and differentially affect the aggressive proclivities of their offspring—proclivities that daughters might call upon in adulthood to improve dominance acquisition and maintenance and, ultimately, reproductive success. Androgens thus appear to underlie female sexual selection in meerkats.

As a final note, although gestational effects can be evidenced in daughters (owing to maternal or fraternal androgens, stressors or early life adversity, endocrine disruptors or enzymatic deficiencies, etc.), they may not necessarily transmit to subsequent generations. Specific perpetuation of masculinised traits in females may owe to genetic mechanisms[27,42,46,50,54] or to an additional androgen-mediated, epigenetic mechanism operating during female sexual differentiation[59]. The latter has been suggested for the highly masculinised, female spotted hyaena[31]. Notably, maternal androgens can change the cellular processes within developing foetal ovaries, altering the ratio of foetal ovarian follicles to androstenedione-producing ovarian stromal cells, thereby increasing the daughter's ability to later raise her own androgen concentrations and, in turn, masculinise both the physiology and behaviour of granddaughters. Together, these androgen-mediated organisational and epigenetic mechanisms could ensure differential acquisition of masculinised phenotypes across generations[31]. Although the latter possibility in meerkats requires confirmation, enduring intergenerational female competitiveness may provide one pathway to the evolution of cooperative breeding.

## Methods

**Study population.** We studied wild meerkats at the Kuruman River Reserve (a ~63 km² area comprising dry riverbeds, herbaceous flats and grassy dunes) in the Kalahari region of South Africa (26°58′S, 21°49′E)[28,48]. Our study period (Nov 2011–Apr 2015) included an extended drought, during which female reproductive success tracked rainfall[22] (Supplementary Fig. 1). The annual mean population size was 270 animals, in 22 established clans of 4–39 animals[15,22]. Habituated to close observation (<2 m), handling and routine weighing[28], the animals are micro-chipped and individually identifiable via unique dye marks, and matriarchs are radio-collared to allow tracking clans. Matriarchs have a distinct morphotype and are distinguishable by weight advantage, altered body proportions, increased reproductive output and older age[14,60,61]. Status, health, life history variables and clan composition are known and routinely monitored, as are pregnancies, detectable at 3–4 weeks of gestation. We used palpation, weight gain and shape change to determine gestational stage in real time, and later confirmed estimated conceptions by backdating 70 days (the duration of meerkat gestation) from the known dates of birth[22,28].

**Focal subjects and datasets.** Five of our 57 unique focal dams changed status during the study, producing 31 dominant dams (a subset of which we treated with antiandrogens, see below) and 31 subordinate dams that carried 89 pregnancies to

**Table 2 Metadata on the focal behavioural observations of meerkat dams.**

| Dam category | Dam features | | | | Total observations | | n focals by pregnancy stage | | | n focals by location and collection period | | | |
|---|---|---|---|---|---|---|---|---|---|---|---|---|---|
| | n | Mean age (year) | Mean clan size | n litters | Hrs | n focals | MP | LP | PP | At den | Whilst ranging | AM | PM |
| DC[a] | 22 | 5.0 | 14.4 | 32 | 234 | 895[a] | 166 | 505 | 224 | 214 | 681 | 504 | 391 |
| Sub$_{DC}$ | 16 | 2.2 | 13.7 | 16 | 74 | 279 | 68 | 173 | 38 | 54 | 225 | 150 | 129 |
| DT[a] | 10 | 5.2 | 15.2 | 10 | 87 | 315[a] | – | 272 | 43 | 65 | 250 | 174 | 141 |
| Sub$_{DT}$ | 6 | 1.9 | 16.0 | 6 | 28 | 109 | 23 | 50 | 36 | 20 | 89 | 57 | 52 |
| Total unique | 37 | 3.6 | 14.8 | 48 | 422 | 1598 | 257 | 1000 | 341 | 353 | 1245 | 885 | 713 |

*DC* dominant control, *Sub$_{DC}$* subordinate control under a DC, *DT* dominant treated, *Sub$_{DT}$* subordinate control under a DT, *MP* mid pregnancy, *LP* late pregnancy, *PP* early postpartum.
[a]Behavioural data obtained from dominant dams prior to flutamide treatment contributed to dominant control data.

term, all with live births within the clan (i.e., excluding abortions and evictions[22]). Given field complexities, dams contributed differentially to endocrine and behavioural datasets across pregnancy stages. Endocrine data derived from 35 individuals contributing a total of 49 serum samples (~1 blood sample/pregnancy) and 36 individuals contributing a total of 156 faecal samples, with 18 individuals contributing both types of biological samples (Table 1). Adult behavioural data derived from 37 unique individuals during 48 pregnancies, observed for 1598 focal sessions, totalling 422 h (Table 2). The focal offspring were 103 unique individuals from 32 litters, representing 14 dominant control (DC), 13 subordinate control (SC) and 5 dominant treated (DT) litters (Table 3). Offspring behavioural data from 5661 focal sessions totalled 1362 h of observation.

**Biological sample collection and processing.** For blood sampling, we individually captured a target DC, SC or DT female upon emergence from the underground den, gently picked her up by the base of her tail, placed her into a cotton sack and anaesthetised her with 4% isoflurane (Isofor; Safe Line Pharmaceuticals, Johannesburg, South Africa) in oxygen, administered via a mask attached to a vehicle-mounted vaporiser. Once fully sedated, we reduced isoflurane dosage to 1–2% and used a 25G needle and syringe to draw 0.2–2 ml of blood from the jugular vein. We immediately transferred the blood to a serum separator tube (Vacutainer®, Becton Dickinson, Franklin Lakes, NJ, USA), allowing it to clot at ambient temperature. We then performed our additional procedures, such as health monitoring, as well as our antiandrogen treatment of DT dams, as described below (all capture, anaesthetic, monitoring and sampling methods across subjects were identical). Once recovered from anaesthesia (~20–30 min post-capture), the female was returned to her clan and closely monitored, including by veterinarians, throughout that and the following day (during this time, behavioural data collection on treated females was suspended, see below). We later centrifuged the blood (500 × g for 10 min at 24 °C) and stored the decanted serum, on site, at minimally −20 °C.

We also opportunistically collected, into clean plastic bags, roughly half the amount of any freshly voided faeces (to avoid disrupting faecal scent marking), immediately placed samples on ice and later stored them at minimally −20 °C. We transported all serum and faecal samples, on ice, to Duke University (Durham, North Carolina) and stored them at −80 °C. Within a year of collection, we lyophilised, pulverised and sifted faecal samples into fine powder and extracted steroid metabolites[15,22]. We weighed 0.2 g of dry powder, mixed it with 2 ml of 90% methanol, placed the mixture on a rotating shaker (30 min), centrifuged it twice, discarding the sediment each time, and stored the methanol-extracts at −80 °C until analysis.

**Endocrine assays.** To measure serum androgen concentrations, we used competitive enzyme immunoassay (EIA) kits (ALPCO diagnostics, Salem, NH, USA), previously validated in meerkats (Supplementary Methods)[15,43]. The serum androstenedione assay has a sensitivity of 0.04 ng/ml using a 25 μl dose, with intra-assay and inter-assay coefficients of variation (CVs) of 5.23% and 8.7%, respectively. The serum testosterone assay has a sensitivity of 0.02 ng/ml using a 50 μl dose, with intra-assay and inter-assay CVs of 7.9% and 7.3%, respectively. We also assayed fAm via EIA (Supplementary Methods). The fAm assay has a sensitivity of 0.2–12.5 ng/ml per plate, with intra-assay and inter-assay CVs of 7.7% and 6.2%, respectively.

To measure fGCm concentrations, we used the ImmuChem double-antibody 125I radioimmunoassay kit for corticosterone (MP Biomedicals, Irvine, CA), also validated in various vertebrates, including meerkats (Supplementary Methods)[22]. The inter-assay CVs were 10.93% and 9.05% for low and high faecal pools, respectively. Intra-assay CV was 5.55%. All individual sample CVs were below 15%.

**Behavioural observations.** We tracked and observed clans minimally every three days; all observation protocols limit human presence to several days/week. For candidate dams, deemed healthy and living in stable clans, we began behavioural

observations after confirming pregnancy, continuing throughout gestation and PP. For offspring, we began behavioural observations when pups were ~1 month old (when they leave the natal den and reliably join the foraging clan). We used focal-observation protocols[62], as these are less vulnerable to detection bias than are critical-incident protocols and allowed us to capture detailed or cryptic behaviour, including across seasons when vegetation can differentially obscure the animals.

We observed focal subjects, in randomised order, during two daily periods, one beginning in early morning, after emergence, until midday (AM), the other resuming late afternoon until the animals retired underground (PM). We recorded behaviour in real time, using CyberTracker software (version 3.263, CyberTracker Conservation) uploaded to handheld palm pilots (Palm T|X, Palm, Inc.), and paused recording whenever the subject was out of view. We tailored our protocols to the meerkats' major activity patterns at different times and locales: 5 min den focals occurred at either end of the day, during the meerkats' brief periods spent above ground at the den, whilst mostly sedentary and prosocial; 30 min ranging focals occurred in the interim, whilst meerkats primarily foraged throughout their territory. The shorter focals at the den allowed us to rotate through our subjects minimally once prior to the clan's departure from or descent into the den.

For adults, we present data on 1) aggression (including competition over an acquired food item and other high-intensity aggression or 'HIA'; the latter comprises the former), 2) scent marking, 3) prosociality, 4) submission and 5) nearest-neighbour (N-N) associations (as a proxy of an ego network; Supplementary Methods). For categories 1, 3 and 4, we recorded the partner and directionality (i.e., initiator versus recipient). For category 5, every 2–3 min or whenever partners changed, we recorded the identities of meerkats within 10 cm or 2 m of the focal dam (for den and ranging focals, respectively). We also report on the frequency and duration of non-target or neutral behaviour performed by all meerkats whilst ranging, including 6) vigilance, a form of cooperative behaviour[44], and 7) the duration of digging for food. For offspring, we present on category 1 (with the addition of targeted begging). We omitted from analysis any focal in which a disrupting interclan interaction occurred. We assessed intra- and inter-observer reliability, with scores exceeding 87% (see Supplementary Methods for additional details on reliability). For the much longer span of offspring observations, we also confirmed that including observer identity as a random variable in the analyses had no significant impact on the findings.

**Selecting controls.** We planned our experimental manipulation, using steadily dissolving antiandrogen pellets that require no removal, in dominant dams only, as their reproductive success is most reliable. Local permitting required that we avoid implanting sham pellets in matriarchs—the key individual per clan—to minimise the number of clans potentially affected by our protocols (i.e., to facilitate uninterrupted, long-term collection of normative life-history data by other researchers). Therefore, we ran multiple alternative controls, the first of which involved a substitute manipulation (Feb 2011–Jan 2012) in three groups of adult, subordinate, male meerkats: a flutamide-treated (~15 mg/kg/day) group that received 1 pellet (300 mg, Innovative Research of America, Sarasota, FL), a sham-treated group that received 1 sham pellet of the same matrix, and untreated controls that received no pellet[43] (see below). Males are ideal for delineating androgen-mediated traits as a basis for assessing flutamide effects in females[19,39,41,42,46].

We then supplemented this sham control in males with several other controls in females (see below), that allowed us to test multiple predictions. Notably, we compared the following: androgen-mediated vs. nonandrogen-mediated behaviour in DT dams, health measures (via abortions, stress hormones and weight) in DC vs. DT dams; indirect behavioural effects in SC dams cohabiting with DC vs. DT dams; match-paired effects on social centrality (via nearest-neighbour analyses) in DC vs. DT dams and; the behaviour of offspring from DC vs. DT and DT vs. SC dams.

**Antiandrogen treatment.** We brought in full-time veterinarians, licensed in South Africa, to perform the minimally invasive implants and monitor the animals before, during and after treatment. Previously, in males[43], we found (a) no negative

**Table 3 Metadata for the focal behavioural observations of meerkat offspring.**

| Maternal category | Offspring features | | | | Total observations | | n focals by age (months) | | | | | | n focals by location and collection period | | | |
|---|---|---|---|---|---|---|---|---|---|---|---|---|---|---|---|---|
| | n | Sex ratio (F/M/U)[a] | Mean clan size | n litters | Hr | n focals | 1 | 2 | 3 | 4 | 5 | 6 | At den | Whilst ranging | AM | PM |
| DC | 46 | (20/25/1) | 16.5 | 14 | 509 | 2199 | 340 | 409 | 426 | 311 | 371 | 342 | 721 | 1478[b] | 1307 | 892 |
| SC | 42 | (19/23/0) | 18.1 | 13 | 548 | 2231 | 207 | 504 | 428 | 415 | 328 | 349 | 547 | 1684 | 1359 | 872 |
| DT | 15 | (8/7/0) | 22.5 | 5[c] | 305 | 1231 | 182 | 171 | 231 | 259 | 220 | 168 | 341 | 890 | 733 | 498 |
| Total unique | 103 | (47/55/1) | 18.5 | 32 | 1362 | 5661 | 729 | 1084 | 1085 | 985 | 919 | 859 | 1609 | 4052[b] | 3399 | 2262 |

DC dominant control, SC subordinate control, DT dominant treated, F female, M male, U unassigned for pups that died prior to first capture.
[a]Offspring number and litter sex ratio at emergence represent starting sample sizes, with subsequent losses over time owing to natural mortality.
[b]Includes two additional observations classified as transitory between being at the den and ranging.
[c]Of an initial 10 litters born to DT dams that carried to term (see Supplementary Fig. 4), five litters (50%) remained for behavioural testing. The other five litters either were lost pre-emergence, owing to observed infanticide by subordinates (1.5 litters), snake predation (1 litter) or unknown causes (1 litter), or were lost shortly after emergence, owing to death (0.5 litter) or taming by farmers (1 litter).

health effects in any of the three groups, (b) no behavioural differences between sham and control groups, (c) the same significant differences between the flutamide-treated group and each of the two control groups, (d) the specifically predicted changes in behaviour owing to androgen receptor blockade and (e) consistency of androgen-specific inhibition, as obtained in other flutamide studies, regardless of administration mode (i.e., oral, injection, topical, subcutaneous implant)[34,39–42]. Neither the procedures associated with minor incision nor the pellet matrix significantly impacted male meerkats[43].

Here, following this male validation, we treated 11 pregnancies from 10 dominant dams with flutamide, at the same dose used in males (~15 mg/kg/day), targeting the estimated last 21 days (LP) of their 70-day gestation (Supplementary Fig. 3). During routine capture, the veterinarian provided each experimental animal with a subcutaneous injection of a non-steroidal, anti-inflammatory painkiller (either 0.2–0.3 mg/kg meloxicam, Metacam, Boehringer or 0.1 ml of metacam, plus 0.1 ml of lentrax or other long-lasting penicillin, depending on availability). Under sterile procedures, the veterinarian used a scalpel to make a 1–2 cm dorsal skin incision between the shoulder blades, blunt dissection to create a small subcutaneous pocket, forceps to insert two 21-day release flutamide pellets (because dominant females are twice the weight of adult, subordinate males[43]), dissolvable material (Vicryl) to suture the skin incision and surgical glue to cover the suture for added protection.

Recovery and release of all dams was as described above (see biological sample collection); however, treated females were monitored more extensively and for longer periods than is routine. Again, no adverse effects of treatment were detected; dominant females maintained their status and continued to fill their matriarchal role (as expected for this short treatment window). Whereas 21% of the contemporaneous dominant controls aborted[22], only one treated female (9%) aborted and that pregnancy was thus excluded from the study. Given costs of natural androgens to dominant female meerkats[57,58], flutamide-treated dams had potentially less health risk than their control counterparts. A few subjects received antibiotic ointment to treat minor infections at the suture site, but these typically occurred after parturition (i.e., after the cessation of focal observations) and had no noticeable effects on behaviour. Owing to difficulties assessing pregnancy stage in real time, particularly under drought conditions[22], treatment periods ultimately extended into PP, so we matched the LP + PP time span in controls (Supplementary Fig. 3). Overshooting treatment was preferable to undershooting, given that the critical period of offspring behavioural masculinisation could extend perinatally, through a combination of androgen transfer via the placenta and maternal milk.

**Analysis of endocrine data**. To examine the relationship between androgens and status in control females across pregnancy, we ran three models in R (version 3.4.3[63]). We analysed serum concentrations using GLMMs in the MASS package (version 7.3-47[64]). We used a Gamma error distribution and log link function, accounting for females sampled across multiple pregnancies by including individual identity as a random factor. We analysed faecal concentrations using LMMs in the lme4 package (version 1.1-21[65]). After log transformation, the response variables conformed to the normal distribution, so we used a Gaussian error distribution with an identity link function. Because we obtained multiple faecal samples per pregnancy, we included litter identity nested within individual as a random factor.

To test if androgen concentrations changed naturally with pregnancy stage, we included the interaction between status (dominant, subordinate) and stage (EP, MP, LP, PP) as fixed factors in the full models, as well as female age (continuous in years; meerkat age does not covary with status for endocrine data) and total monthly rainfall (continuous in mm$^2$; rainfall highly correlates with year and territory, but was the most sensitive measure of temporal quality). Owing to drought conditions, we excluded clan identity from most analyses, because membership and location changed too often between pregnancies to be a reliable measure. Because we obtained all serum samples in the morning, we included collection period (AM or PM) only as a covariate in the fAm model (Supplementary Table 1).

Each full model included all probable independent terms and biologically relevant interactions; we then obtained minimal models by sequentially removing the least significant factors ($P < 0.05$), starting with two-way interactions. We confirmed validity of the final models (full and minimal) using a forward stepwise procedure[66] and verified our assumptions by checking residuals for normality and homogeneity of variance. We assessed collinearity between main effects via variance inflation factors (VIFs), all of which were <2, in the R package "car" (version 2.1-6[67]). We determined significance of fixed factors through maximum likelihood estimation and likelihood ratio tests following a $\chi^2$ distribution. Significant interactions and pairs from three-level factors (e.g., EP, MP, LP or MP, LP, PP) were compared to each other using post hoc pairwise comparisons (LSD) in the lsmeans package (version 2.30-0[68]). Statistical tests were two-tailed throughout and, unless otherwise stated, we present means and standard errors.

Lastly, we analysed the effect of antiandrogen treatment on the concentrations of androgens and stress hormones in dominant dams. Because treated females did not contribute serum samples and because faecal sampling is noninvasive and better captures natural changes in the stress response, we addressed this question using faecal samples. We ran new models (comparable to the prior fAm model, above) to test if flutamide influenced the fAm or fGCm concentrations of treated versus control

dominant dams. The full models included experimental condition (control, treatment), age, total monthly rainfall and collection period. Additionally, we included pregnancy stage (LP, PP) as a random factor to control for the timing of treatment.

**Analysis of adult life-history data**. We tested if dam age, weight or clan size related to treatment condition, using LMs in the stats package (version 3.6.1) in R (version 3.6.1[63]), for which we included the dominant female's treatment condition as the fixed factor. A first analysis of values at conception served to verify that we had randomly assigned treatments. Indeed, there were no differences by age ($t_{1,26} = 0.45$, $P = 0.655$), weight ($t_{1,27} = 1.34$, $P = 0.191$) or clan size ($t_{1,29} = 0.52$, $P = 0.609$) between dominant control and treated females. A second analysis of postpartum weight served to confirm the lack of negative health effects of flutamide treatment, as would be revealed by weight loss (see "Results" section). To control for pre-treatment condition, this analysis also included weight at conception as a covariate.

**Analysis of observed adult behaviour**. To test our behavioural predictions for dams by 1) status, 2) treatment condition (for dominant females), or 3) matriarch's treatment condition (for subordinate females), we analysed rates of aggression, scent marking, prosociality and submission using zero-inflated GLMMs in the glmmADMB package (version 0.8.3.3[69]) in R (version 3.4.3[63]). We focused on the MP-PP increase in behaviour, rather than the potential PP decrease, because behavioural changes often lag behind endocrine changes. We used either a Poisson or negative binomial error distribution, and used the distribution with the lowest Akaike's Information Criterion (AIC) value for subsequent model selection, with focal duration as an offset. To examine the effects of antiandrogen treatment on non-targeted behaviour, including rates of digging and vigilance, we used models for which focal duration was included as an offset. For duration measures, we used an LMM with a gaussian (and a log link) error distribution for digging, and a GLMM with zero-inflated Gamma (and a log link) distribution for vigilance, using the glmmTMB package (version 1.0.2.1[70]) in R (version 3.6.1[63]). To account for repeated sampling (of individuals and pregnancies), we included litter nested within individual as a random factor, unless convergence failed, in which case we included only litter as a random factor, as this was the more sensitive measure.

First, we tested if normative behaviour (save submission, the direction of which depends on status) differed by status as the unmanipulated pregnancies progressed. We included the interaction between status and pregnancy stage (MP, LP, PP) in each full model, as well as clan size (continuous in number of members present during observation), total monthly rainfall and collection period. For behaviour that occurred irrespective of location, such as aggression (initiated or received) and scent marking, we included the type of focal (at den, whilst ranging) as a covariate. We did not do so for food competition (which occurred primarily whilst ranging) or prosociality and submission (which occurred primarily at the den). Owing to collinearity between age and status when comparing behaviour across classes, as assessed via VIFs (all other variables >2), we did not include age in these models (Supplementary Table 2).

Second, we tested if flutamide directly affected the behaviour of treated dominant dams, relative to control dominant dams (Supplementary Table 3), and third, if flutamide indirectly altered the behaviour of cohabiting subordinate dams, for which we compared the contemporaneous behaviour of subordinate dams whose matriarch was treated to that of subordinate dams whose matriarch was left untreated (Supplementary Table 6). In both cases (involving within-class comparisons), we included the dominant female's treatment condition as a fixed factor in each full model, as well as the respective dominant or subordinate female's age, clan size, total monthly rainfall and collection period. The type of focal was included as a covariate whenever relevant. Additionally, pregnancy stage for focal dominant (LP, PP) or subordinate (MP, LP, PP) dams was included as a random factor to control for potential gestational differences. Because the behaviour of subordinate dams remained consistent across pregnancy (Fig. 1d–f), we increased their sample size by extending this analysis to include subordinates at MP.

As in our endocrine analyses, we initially included all probable independent terms and biologically relevant interactions in the full models. A minimal model was obtained by sequentially removing terms based on AIC, with validity of the final models being confirmed using a forward stepwise procedure[66]. Significance of fixed factors was determined through maximum likelihood estimation and likelihood ratio tests following a $\chi^2$ distribution, and significant interactions and three-level factors were compared using post hoc pairwise comparisons (LSD) in the lsmeans package (version 2.30-0[68]).

**Analysis of inferred evictions**. We tried several ways to analyse the number of evictions by dominant control versus treated females during LP (for $n = 31$ pregnancies) using changes in clan composition. First, we excluded the pregnancies of dominant females when there were no other adult females present. Then, we accounted for differences in clan size, the number of adult females present (i.e., the number of potential eviction victims), the number of times the same female was repeatedly evicted, and the timing for the onset of treatment (i.e., if victims had already been evicted when dominant females started treatment late). We selected the mean percent of adult females evicted (out of the total number of adult females residing in the clan); nevertheless, accounting for these important variables for so

few occurrences of evictions by treated matriarchs reduced the power necessary to properly model this question.

**Analysis of nearest-neighbour (N-N) associations**. To determine if antiandrogen treatment altered the matriarch's social relationships (or centrality), we calculated dyadic rates of N-N associations (using scan sampling) for dominant dams and their adult clan members, during both treatment and control periods. We restricted this analysis to the four dominant females observed for more than an hour during both a treated and control pregnancy (totalling eight pregnancies); however, the sample size, based on Satterthwaite's approximation method, reflects the number of adults associating with these dominant dams. The resultant dataset contained 110 dyads, including 43 adult female and 67 adult male partners (27 and 40 of which, respectively, were unique), representing 77 total hours of focal observation. Dominant females averaged 1.83 N-N scans per adult clan member per hour across periods (S.D. = 1.20, range = 0–6.35). We ran an LMM in R (version 3.6.1[63]), using the lme4 package (version 1.1-21[65]), with dyadic rates of N-N association as the outcome variable and partner sex, condition (treatment versus control), adult clan size, and indicator variables for dominant males and subordinate dams as main effects, and dominant female identity and partner identity as random effects. We calculated VIFs using the R package "car" (version 3.0-5[67]); all VIFs were <1.2, indicating adequate independence of predictor terms (Supplementary Table 5).

**Analysis of observed offspring behaviour**. We used a combined model to examine the offspring's rates of initiated aggression by maternal condition, including status (dominant versus subordinate control) and treatment (dominant control versus treated; Supplementary Table 6). We used GLMMs with a negative binomial error distribution in the glmmTMB package (version 1.0.1[70]) in R (version 3.6.3[63]) to account for overdispersion and zero-inflation, with focal duration as an offset. To account for repeated sampling, we included individual nested within litter nested within dam as a random factor, unless convergence failed, in which case we included only individual nested within litter, as this was a more sensitive measure. As in our analyses of adults, we included all probable independent terms (i.e., maternal condition, offspring sex, offspring age in months, clan size, rainfall, focal location and collection period) and biologically relevant interactions (i.e., between maternal condition and offspring age, and between clan size and rainfall) in the full model. We obtained a minimal model by sequentially removing terms based on AIC and confirmed final-model validity using a forward stepwise procedure[65]. Fixed factor significance was determined through maximum likelihood estimation and likelihood ratio tests following a $\chi^2$ distribution, and significant interactions and three-level factors were compared using post hoc pairwise comparisons (LSD) in the lsmeans package (version 2.30-0[67]).

**Reporting summary**. Further information on research design is available in the Nature Research Reporting Summary linked to this article.

## Data availability

Source data are provided with this paper. The data that support these findings are available at https://github.com/cls83211/dreaetal2021. All data generated in this study has been deposited in the GitHub database dreaetal2021 under accession code: https://doi.org/10.5281/zenodo.564382571. The raw data have been anonymized, but are all included in the database. The source data generated for figures in this study are provided in the Source Data file. Source data are provided with this paper.

## Code availability

The code that support these findings are available at https://github.com/cls83211/dreaetal2021. All R code generated in this study has been deposited in the GitHub database dreaetal2021 under accession code: https://doi.org/10.5281/zenodo.564382571.

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

## Acknowledgements

We thank the Kalahari Research Trust, Northern Cape Conservation Authority and Kotze family for permission to work at the Kalahari Meerkat Project (KMP). We are grateful to B. Ashton, M. Böddeker, S. Cox, L. Garside, V. Goerlich-Jansson, I. Goncalves, E. Kabay, D. Pfefferle, A. Reyes, E. Terrade, M. Thavarajah and D. Walsh for help with data and sample collection; S. Bischoff-Mattson, D. Gaynor, L. Howell, M. Manser, L. Marris, J. Samson, M. Siekelova, N. Thavarajah and the KMP volunteers for logistical support; J. Drewe, M. Lategan, J. Maud, S. Patterson and C. Visser for veterinary care; I. Chandra, S. Epstein, M. Horowitz, and C. Southworth for assistance with behavioral data curation; J. delBarco-Trillo for data sharing; and the Statistics Consulting Center at Duke University. This research was funded by the National Science Foundation (IOS-1021633 to C.M.D.). We relied on records maintained by the KMP, which has been supported by European Research Council Grant (No 294494 to T.C.-B.) and Swiss National Science Foundation Grant (31003A 13676 to M. Manser). Cambridge, Duke, and Zurich Universities supported the KMP during the span of this study.

## Author contributions

C.M.D. conceived, designed, funded and supervised the study, with input from T.C.-B., who also contributed access to the meerkat population and life-history records. C.S.D., L.K.G. and J.M. managed the field project and animal captures and, with D.V.B. and K.N.S.-K., collected samples and behavioral data and, with K.A.D.-S. and C.L.S., curated the data. C.S.D., L.K.G., K.A.D.-S. and K.N.S.-K. performed the extractions and endocrine assays. C.S.D., C.L.S. and J.T.F. analysed the data, prepared the figures, tables and source datafiles, and with C.M.D. wrote the methods. C.M.D. wrote the manuscript with contributions from C.S.D., C.L.S. and J.T.F. and with edits from L.K.G. and T.C.-B.

## Ethics approval and consent to participate

Our protocols were approved by and carried out in accordance with the Institutional Animal Care and Use Committee of Duke University (Protocol Registry Numbers A171-09-06 and A143-12-05) and the University of Pretoria's Animal Use and Care Committee (Ethical Approval Number EC074-11).

## Competing interests

The authors declare no competing interests.
