## [Peer Review File · Nature Communications]

An intergenerational androgenic mechanism of female intrasexual competition in cooperatively breeding meerkatsREVIEWER COMMENTS

Reviewer #1 (Remarks to the Author):

In this ms Drea and colleagues first describe androgen profiles of dominant and subordinate female meerkats in relation to pregnancy status as well as agonistic behaviour of mothers and offspring and prosocial behaviour received by other group members. Basically, dominant females have higher androgen concentrations during most of pregnancy and they and their offspring are more aggressive than subordinate females or their offspring. Drea et al. then manipulate androgen profiles of dominant females during late pregnancy and observe the associated changes in behaviour of females and their offspring, which is consistent with an effect of androgens on these behaviours.

The endocrine profiles of dominant and subordinate female meerkats are impressive, especially given that all of this has been collected in the field. Unfortunately, I am less enthusiastic about other parts of the manuscript, in particular with regard to the sloppy use of citations and the experimental procedures.

First, I got really confused when I tried to look up references. For instance, to look up behavioural observation protocols and experimental procedures on hormone manipulations the authors refer to reference 35, which is not on meerkats and does not describe these procedures. Unfortunately, this is not the only case. A good number of citations are simply wrong, i.e. they refer to papers other than the context provided in the main text (e.g. refs 35-40 do not fit at all in none of the contexts they are mentioned). In addition, I got the impression that in some places the citations are biased, i.e. the authors selectively cite only those studies that support their view, but ignore other studies with a contrasting view point. Together, the inaccuracy and selectiveness in how citations are used throughout the paper does not help to increase my overall confidence in the manuscript.

Second, and most importantly, it seems the treatment groups consisted of a group of dominant female meerkats treated with flutamide, that is they had to be captured, sedated, a small incision to be made to insert the implant, etc. and a control group that was not captured and treated at all. I understand the authors wanted to limit the number of clans affected by the protocol, but from an experimental point of view the data are biased, because the treatment variable is confounded and there is no proper control. To demonstrate that sham-implanted and untreated control males are the same is not a sufficient replacement, as males and females can respond differently to the same kind of treatments. In the end, the authors cannot distinguish whether the behavioural effects of the treatment are due to the androgen suppressing effect of flutamide or due to capturing, sedating and implanting an agent (or a potential infection related to the implantation). While I am fully aware of the challenges of conducting such an experiment with free-living animals and applaud the authors in doing so, I think a proper control group would have been necessary, especially if attempting to publish the findings in a high-profile journal such as Nature Communications.

The authors often refer to spotted hyenas. In that context it would be good to learn more about rank inheritance in meerkats and whether the mechanisms that have been suggested for hyenas (one publication suggesting a mainly hormonal effect another one a behavioural effect (Dloniak et al., 2006; East et al., 2009)).

Specific comments:

line 54 "raised androgen concentrations relative to conspecific males" and the citations (refs 13,14) ignore the fact that free-living reproductively active male spotted hyenas (immigrant males) have higher levels of androgens than females (Goymann et al., 2001). Ref 13 did only measure female androgens without comparing it to males. And Ref 14 did not measure circulating androgens, but androgen metabolites. However, the kind of androgen metabolite assay that has been used by Ref 14 is likely to measure other substances than androgen metabolites (Pribbenow et al., 2017) and therefore cannot be considered a reliable assay for the comparison of female and male androgen concentrations of spotted hyenas. The best available evidence is therefore (Goymann et al., 2001) showing that male testosterone concentrations are on average 13+ times higher than those of females, whereas androstenedione concentrations of females are 1.4 times

higher than those of males. Overall, this is little evidence of "raised androgen concentrations relative to conspecific males" in spotted hyenas.

Line 84f: you mention that daughters of selected species (and you cite hyenas, lemurs and tree swallows, refs 14,30,31,33) might be differentially advantaged to later compete for dominant status, but I do not think that most of the citations provide any evidence for this.

Line 92ff: isn't it a general mammalian pattern that androgens increase during gestation?

Line 99: another component would have been to elevate androgens in pregnant subordinate females. These don't happen often enough?

Line 128ff: it seems aggression did not increase in dominant females during gestation, so androgens mainly correspond to food competition and prosocial behaviour received

Line 144ff: more details on the kind of behaviours that were recorded in cubs would have been necessary. Also, would be great to include original data in Fig. 2 and explain the y-axis. I do not understand what "frequency/mean focal duration or 14.4 min" means. Frequency as a measure of number of initiated aggression per observation period would make sense, but also a list of behaviours that were scored as an initiation of aggression would have been required. Also, is this controlled for the number of potential interaction partners?

Line 144ff/207ff: any idea what the role of aggression between cubs is until month 6? After that they seem to be similar regardless of rank. In that context, as mentioned above, would be good to get more info on the role of maternal rank and cub aggression for survival and fitness of offspring...

Line 234: since the reference does not seem to be correct, which reversed effects of androgen supplemented females are you talking about?

Line 251: here and above most citations are out of place, but one that demonstrates the effect of the mother best would probably be East et al. (2009), demonstrating with natural cross-fostering that it is basically maternal rank that is important for status acquisition of cubs, i.e. cubs of low-ranking females cross-fostered by high-ranking females obtain high social status and vice versa. This effect seems to be much larger than the tiny differences in androgen-related agonistic acts of cubs reported by Dloniak et al. (2006).

Line 286ff: is there actually evidence for this in meerkats?

Line 291ff: Not sure the mechanism described here and referred to Ref 30 is still accurate, i.e. I recommend to check the work by Ned Place, Steve Glickman and coworkers who suggest large genetic effects, i.e. (Glickman et al., 2006; Place et al., 2002).

References

- Dloniak SM, French JA, Holekamp KE, 2006. Rank-related maternal effects of androgens on behaviour in wild spotted hyenas. *Nature* 440:1190-1193.
- East ML, Honer OP, Wachter B, Wilhelm K, Burke T, Hofer H, 2009. Maternal effects on offspring social status in spotted hyenas. *Behav Ecol* 20:478-483. doi: 10.1093/beheco/arp020.
- Glickman SE, Cunha GR, Drea CM, Conley AJ, Place NJ, 2006. Mammalian sexual differentiation: lessons from the spotted hyena. *Trends in Endocrinology & Metabolism* 17:349-356.
- Goymann W, East ML, Hofer H, 2001. Androgens and the role of female "hyperaggressiveness" in spotted hyenas (*Crocuta crocuta*). *Horm Behav* 39:83-92.
- Place NJ, Holekamp KE, Sisk CL, Weldele ML, Coscia EM, Drea CM, Glickman SE, 2002. Effects of prenatal treatment with antiandrogens on luteinising hormone secretion and sex steroid concentrations in adult spotted hyenas, *Crocuta crocuta*. *Biol Reprod* 67:1405-1413.
- Pribbenow S, Shrivastav TG, Dehnhard M, 2017. Measuring fecal testosterone metabolites in spotted hyenas: Choosing the wrong assay may lead to erroneous results. *J Immunoassay Immunochem* 38:308-321. doi: 10.1080/15321819.2016.1260584.

Reviewer #2 (Remarks to the Author):

The goal of this manuscript is to test an hypothesis suggesting that androgens mediate agonistic interactions among female meerkats as a form of intrasexual competition. The authors combine data on naturally-occurring behavior and hormone concentrations with data obtained from field experiments where several dominant females were treated with anti-androgens (AA) during late pregnancy and early during the postpartum period to assess both activational and organizational effects of androgens on behaviors that function importantly in female meerkat breeding competition, and thus effectively mediate reproductive monopoly in this species. The work is beautifully grounded in the framework developed by Phoenix et al (1959). The authors managed to obtain surprisingly (and delightfully) large sample sizes with respect to their behavioral observations and their sampling of hormone concentrations in both serum and fecal matter. The logic linking their hypothesis with expected outcomes seems very sound, the quality of the data is very high, and the statistical analyses all seem sound. I also found the metadata tables to be very helpful, so I was glad the authors decided to include those in the main manuscript. The writing is generally clear and succinct, and the author's interpretations of the data seem very reasonable with the exception of the "heritable" component discussed below. The data presented in the Supplementary Materials help rule out various alternative interpretations of the main findings to those on which the authors focus. For instance, although to minimize the impact of this experiment on the study population, dominant control females were not implanted with sham pellets, I found the other types of controls applied (implanting males with both experimental and control pellets, comparing treated females to themselves when untreated, controls applied in the statistical analysis of the data, etc.) to be very convincing. Furthermore, although it is conceivable that the trivial surgical procedure itself uniquely altered the behavior of treated females, this cannot explain the observed changes in the behavior of group-mates documented in the tables in the Supplementary Materials, particularly the behaviors not involving direct interactions with treated females.

I believe this is an important paper, particularly for its emphasis on a broader role than has been previously recognized for important functions of androgens in mediating reproductive competition among female vertebrates. Whereas the first author has explored the same issues in what many refer to as "sex-role reversed" species such as lemurs and spotted hyenas, the current demonstration of their importance in meerkats exploits a much more tractable study system than would be possible with those other, larger species. Despite some variability in the duration of AA treatments, the tractability of this study system, and the strong and consistent support in their data for their hypothesis, allow the authors to make this paper a real tour de force.

My only substantive concern is that, although the authors claim that the androgenic mechanism they explore here is heritable, and although it might indeed truly be heritable, the authors have not actually shown that here. Instead, they have shown only the existence of intergenerational transmission of 'masculinized' phenotypes in meerkats. Therefore, the authors either need to remove the word "heritable" from the title and elsewhere in the paper or to actually assess the heritability of this trait using the animal model, with methods described by Kruuk (2004), and as employed to assess heritability of aggressiveness in spotted hyenas by Yoshida et al (2016). This should be fairly straightforward, as the Kalahari Meerkat Project should certainly have all the necessary pedigree data available for these calculations. Using the animal model as Kruuk (2004) suggests would also allow the authors to determine what percent of the overall variation in androgen-mediated aggressiveness is due to heritability, to maternal effects, and what percent remains unexplained. Both are mentioned in this manuscript, but the reader is left wondering about this. If the authors cannot use the animal model here for some reason, I believe they need to reduce their claim about this trait being heritable to the status of an untested hypothesis.

Misc. smaller issues:

L 75: "would contribute" should be toned down a bit to say "might contribute".

L 135-136: "...received more prosociality..." is very awkward. I'd be inclined to replace that with something more like "... received more prosocial overtures from clan-mates..."

Legend to Fig 4: Lines 793-795 don't make sense to me. Why the parenthetical reference to Fig 4 here?

In the Results section, the use of "dominants versus subordinates" generates some awkward sentences. Why not say (eg., L 149-150) more straightforwardly that "dominant mothers initiated more aggression than did subordinate mothers." Or in line 154 "...from dominant than subordinate mothers..." (rather than dominant versus subordinate mothers).

L 178: "Treated matriarchs evicted 2/3 fewer subordinate females than did controls": The fraction here should be replaced by a percentage to make this sentence easier to follow.

L 234: Reference 36 is cited here yet this paper contains not a single word about "androgen-supplemented females of other species," which is suggested by this citation. This paper discusses rank acquisition in primates and spotted hyenas, but there was no androgen supplementation mentioned in either taxon.

The discussion of the importance of timing of androgen exposure (lines 258-276) should certainly cite some of the critical work by Kim Wallen on this in macaques.

In Methods, what is the difference between project personnel and non-project personnel, and why is it necessary to make this distinction?

References cited here

Kruuk, L.E.B. (2004). Estimating genetic parameters in natural populations using the "animal model". *Philos. Trans. Roy. Soc. Lond. B: Biol. Sci.* 359: 873-890.

C. H. Phoenix et al. (1959) Organizing action of prenatally administered testosterone propionate on the tissues mediating mating behavior in the female guinea pig. *Endocrinology* 65: 369.

Yoshida, K.C.S., Van Meter, P.E. & Holekamp, K.E. (2016) Variation among free-living spotted hyenas in three personality traits. *Behaviour*. 153: 1665-1722.

Kay E. Holekamp

Reviewer #3 (Remarks to the Author):

I enjoyed reading the manuscript by Drea et al and applaud the attempt to understand both the organizational and activational effects of androgens in female meerkats. The paper is well written and thorough. I am not aware of similar investigations in wildlife. The long-term study system that is composed of many clans is ideal to test the interaction between androgens and dominance in females across generations. The only concern that I have about the study is the lack of validation in females. I appreciate the difficulty involved in validating the procedures on wildlife in the field, especially when part of a long-term research. However, these validations can and should be carried out on captive meerkats.

Other comments:

In order to appreciate the effects of androgens in pregnancy and on the offspring, more details describing the system are needed in the Introduction. For example, how many offspring are there on average in a litter? Do dominant females have more female-biased offspring sex ratios?

Maybe I missed it, but why was the experimental implantation at 3 weeks? How was this time period chosen? Why not 2 or 4 weeks? How long is meerkat pregnancy? Were implants removed following 3 weeks?

In a related matter, since there is variability between individuals in the timing that they received the implant (LP or PP), how was the timing of the implant insertion accounted for in the analysis of

offspring behaviour?

Was year included in the statistic models?

Was pup anogenital distance (AGD) measured when pups were handled? This may provide an indication of earlier androgen exposure and an alternative way to assess in utero masculinization.

Have the authors related serum and fecal androgen levels? Not sure how related they are, and whether the same outcome would be with serum T, A4. Have the authors measured serum androgens as a result of the implants?

The implanting procedure described is invasive and includes pain and anesthesia. Including a control group with empty implants is required in experiments and rules out the possibility that the behavioural effects following insertion are due to the inhibitor rather than pain. I think that such assays also require validation in the same sex as later experimented on, as females are different than males, especially while pregnant.

I found the Discussion short on evolutionary implications. In order to influence thinking in the field, I suggest expanding the Discussion to include additional aspects.

I suggest moving Tables 1-3 to the SI.

Figure 2: what data is the model based on? Adding the actual data on the figure may be helpful.

Table S1: Please add measured concentrations of androgens. It would be interesting to see.

Response to the Reviewers' Comments

Reviewer #1 (Remarks to the Author):

In this ms Drea and colleagues first describe androgen profiles of dominant and subordinate female meerkats in relation to pregnancy status as well as agonistic behaviour of mothers and offspring and prosocial behaviour received by other group members. Basically, dominant females have higher androgen concentrations during most of pregnancy and they and their offspring are more aggressive than subordinate females or their offspring. Drea et al. then manipulate androgen profiles of dominant females during late pregnancy and observe the associated changes in behaviour of females and their offspring, which is consistent with an effect of androgens on these behaviours.

We thank Reviewer 1 for this summary.

The endocrine profiles of dominant and subordinate female meerkats are impressive, especially given that all of this has been collected in the field. Unfortunately, I am less enthusiastic about other parts of the manuscript, in particular with regard to the sloppy use of citations and the experimental procedures.

3. I take full responsibility and apologize for the multiple errors in referencing that caused significant confusion for reviewers. It owed to uploading the wrong version of citations in the transfer of our manuscript from *Nature* to *Nature Communications*, and has been rectified. I also wanted to assure Reviewer 1 that the oversight affecting citation order was an isolated and uncharacteristic incident.

First, I got really confused when I tried to look up references. For instance, to look up behavioural observation protocols and experimental procedures on hormone manipulations the authors refer to reference 35, which is not on meerkats and does not describe these procedures. Unfortunately, this is not the only case. A good number of citations are simply wrong, i.e. they refer to papers other than the context provided in the main text (e.g. refs 35-40 do not fit at all in none of the contexts they are mentioned). In addition, I got the impression that in some places the citations are biased, i.e. the authors selectively cite only those studies that support their view, but ignore other studies with a contrasting view point. Together, the inaccuracy and selectiveness in how citations are used throughout the paper does not help to increase my overall confidence in the manuscript.

4. See Response 3.

Our study is novel, fairly specific, and, to our knowledge, not particularly controversial. We thus do not know to which 'biased' citations the reviewer is alluding. We selected the most pertinent studies (often those serving multiple purposes to accommodate restrictions on citation number), published in reputable journals, by leaders in the field, and would happily address specifics or rectify oversights, should they be identified.

Second, and most importantly, it seems the treatment groups consisted of a group of dominant female meerkats treated with flutamide, that is they had to be captured,

sedated, a small incision to be made to insert the implant, etc. and a control group that was not captured and treated at all.

5. This summation is erroneous. All of the animals were captured and sedated, as originally stated (e.g., lines 487-491) and further punctuated (line 493).

I understand the authors wanted to limit the number of clans affected by the protocol, but from an experimental point of view the data are biased, because the treatment variable is confounded and there is no proper control. To demonstrate that sham-implanted and untreated control males are the same is not a sufficient replacement, as males and females can respond differently to the same kind of treatments. In the end, the authors cannot distinguish whether the behavioural effects of the treatment are due to the androgen suppressing effect of flutamide or due to capturing, sedating and implanting an agent (or a potential infection related to the implantation). While I am fully aware of the challenges of conducting such an experiment with free-living animals and applaud the authors in doing so, I think a proper control group would have been necessary, especially if attempting to publish the findings in a high-profile journal such as Nature Communications.

6. We appreciate Reviewer 1's focus on the appropriateness of the controls, as it is central to the validity of our study. To clarify, the sham limitations we experienced were not our desire, they were imposed (lines 458-463). To handle what otherwise would have become a permitting issue, we delayed the start of our project and instead ran the year-long male study (delBarco-Trillo et al. 2016) to test if sham controls were necessary (lines 463-469; 478-485). As they were not necessary in males, we could proceed with a modified female study, involving multiple *additional* controls (lines 470-476). It is thus not the case that there was 'no proper control'. See also Response 1.

See Response 5 about uniform capture and sedation.

To be precise, flutamide does not suppress androgen concentrations (Fig. S6a; sometimes it raises concentrations owing to feedback loops), it blocks androgens from binding to the androgen receptor. This information had been presented in the methods, but we have now moved it up to the introduction for greater visibility (lines 177-179).

With the necessity of avoiding shams, our inclusion of multiple controls produced a wealth of new information and insight, unparalleled in any other single study on this topic. See Response 2 on the issue of our study's appropriateness for publication in a high-profile journal.

The authors often refer to spotted hyenas. In that context it would be good to learn more about rank inheritance in meerkats and whether the mechanisms that have been suggested for hyenas (one publication suggesting a mainly hormonal effect another one a behavioural effect (Dloniak et al., 2006; East et al., 2009).

7. See Response 2 regarding scope.

We have inserted the word 'matrilineal' (line 302) in relation to hyena society (to better distinguish it from that of meerkats), but otherwise see no call for additional information on spotted hyena behavioral ecology. We reference multiple taxa and species in the context of female social dominance, but unfortunately do not have space to provide more in-depth review of non-study species. Instead, we have provided more natural history information about meerkats (as requested by Reviewer 3; see Response

13) and/or moved meerkat information from the methods to the introduction for more visibility (lines 61-71). Readers can thus make their own assessments about the similarities or differences with respect to their species of interest.

The spotted hyena does feature more prominently than other species in the context of the endocrine mechanism of female aggression, because it is the species in which the mechanism has been best tested and described. That is not to say that we think there is *only* an endocrine mechanism at work (or vice versa).

Given later discussion on the topic of heritability (see Response 11), it is also relevant to note that 'rank inheritance' is incorrect terminology and has created unfortunate perceptions. Notably, rank (a number given to an animal's presumed place in society) is a human construct that misrepresents dominance, which is relational and thus cannot be considered a trait of the individual and cannot be inherited (see Bernstein 1981 *BBS*). Instead, hyena cubs *gradually acquire* an approximation of their mother's current social status through learning, just like many primates do. Their status is subject to change, even if change is infrequent. As a behavioral process of *maternal rank acquisition*, it can be hormonally influenced, so again, these are interactive, not alternative processes.

Lastly, meerkats have a dominance system that differs from that of hyenas (e.g., despotic vs. linear; lines 61-62). Given their cooperative breeding, the maternal role is also minimized (lines 301-306). These factors combine to make the topic suggested by Reviewer 1 less germane for meerkats.

Specific comments:

line 54 "raised androgen concentrations relative to conspecific males" and the citations (refs 13,14) ignore the fact that free-living reproductively active male spotted hyenas (immigrant males) have higher levels of androgens than females (Goymann et al., 2001). Ref 13 did only measure female androgens without comparing it to males. And Ref 14 did not measure circulating androgens, but androgen metabolites. However, the kind of androgen metabolite assay that has been used by Ref 14 is likely to measure other substances than androgen metabolites (Pribbenow et al., 2017) and therefore cannot be considered a reliable assay for the comparison of female and male androgen concentrations of spotted hyenas. The best available evidence is therefore (Goymann et al., 2001) showing that male testosterone concentrations are on average 13+ times higher than those of females, whereas androstenedione concentrations of females are 1.4 times higher than those of males. Overall, this is little evidence of "raised androgen concentrations relative to conspecific males" in spotted hyenas.

8. We apologize that our meaning was not clearer in our initial submission, and have now included additional information to capture the relevant nuances regarding androgen concentrations. We had worded our original sentence for brevity, to encompass the range of patterns observed in all of the different species mentioned and had used the term 'androgens,' which includes both androstenedione and testosterone. The importance of these two hormones in hyenas has long been established in the literature (e.g., Racey & Skinner 1979 *J Zool*) and more recently in other masculinized females (e.g., Grebe et al. 2019). We can see from the confusion generated that we may have been overly brief, as we were not referring only to testosterone and also did not mean

'uniformly greater than.' We have now further parsed our original statement to better qualify the distinctions, both relative to males and between species, and with reference to both key androgens (lines 53-57). That said, as highlighted in yellow above, our original statement was correct: A4 increases are exceptional and not at all 'little' evidence,' including because A4 is the immediate precursor to testosterone and accounts for why testosterone concentrations soar late in spotted hyena pregnancy (Licht et al. 1992), even relative to males. This is the key point. We have thus also mentioned pregnancy in the edited version, even though females of certain species (e.g., hyrax, meerkat) have high androgen concentrations year-round. Any additional discussion about differences in male testosterone concentrations (e.g., resident vs. immature) are off topic (see Response 2). See also Response 4 on references we selected to serve multiple purposes.

Line 84f: you mention that daughters of selected species (and you cite hyenas, lemurs and tree swallows, refs 14,30,31,33) might be differentially advantaged to later compete for dominant status, but I do not think that most of the citations provide any evidence for this.

We indeed cited studies from four different research groups (Licht/Glickman, French/Holekamp, Drea, Schwabl), all leaders in this area and all suggesting the same possible interpretation of their data. We had qualified those suggestions (with e.g., 'may be') and thus see no cause for concern. Unfortunately, without more specifics from the Reviewer we are not able to elaborate further.

Line 92ff: isn't it a general mammalian pattern that androgens increase during gestation?

Yes, it is a general, but weak pattern seen in *early* mammalian gestation. The strong pattern in *late* gestation is seen in the select species mentioned (lines 98-100). See also Response 8 on the relevance of gestational stage.

Line 99: another component would have been to elevate androgens in pregnant subordinate females. These don't happen often enough?

Correct, that would be an alternative approach and we have modified the text to acknowledge the broader range of possible endocrine manipulations (lines 109-111). As Reviewer 1 suspects, this alternative would not be a logistically good one in meerkats. Subordinates are more numerous, but have fewer pregnancies on an individual basis, lose a greater percentage of those pregnancies (see Supplementary Figure 1), and/or are likely to be evicted from the clan when pregnant (and hence lost to field observers). So, manipulation of subordinate pregnancies would require a substantially greater investment (time, effort, finances) with far less return. This is the same reason we did not treat subordinates with antiandrogens.

Line 128ff: it seems aggression did not increase in dominant females during gestation, so androgens mainly correspond to food competition and prosocial behaviour received

That summation is incorrect. Dominant dams initiated more food competition - a key aggressive behavior, as we have now better clarified (lines 144-146), - more intense overall aggression, and more scent marking (lines 144-151).

Line 144ff: more details on the kind of behaviours that were recorded in cubs would have been necessary. Also, would be great to include original data in Fig. 2 and explain the y-axis. I do not understand what “frequency/mean focal duration or 14.4 min” means. Frequency as a measure of number of initiated aggression per observation period would make sense, but also a list of behaviours that were scored as an initiation of aggression would have been required. Also, is this controlled for the number of potential interaction partners?

9. Behavioral details were presented in the methods (under Behavioral Observation), and we have now further elaborated (lines 439-452).

Regarding Figure 2, our sample sizes for offspring behavior were exceptionally large for a mammalian study. We therefore left the main in-text figure as is, but to accommodate this Reviewer’s request (and that of Reviewer 3), we have added a supplementary figure of the raw data as hexagonal density plots (Fig S2).

The confusing part of the legend was a definition of ‘predicted rates’ based on the mean focal duration of 14.4 min. We have changed it to a simple hourly rate for clarity (line 893).

Yes, all analyses of interactive behavior took account of the number of potential partners in interactions (necessary also for e.g., evictions). We have specified that recording clan composition was also part of routine KMP data collection (lines 365-366). We thank Reviewer 1 for drawing our attention to this omission.

Line 144ff/207ff: any idea what the role of aggression between cubs is until month 6? After that they seem to be similar regardless of rank. In that context, as mentioned above, would be good to get more info on the role of maternal rank and cub aggression for survival and fitness of offspring...

Line 234: since the reference does not seem to be correct, which reversed effects of androgen supplemented females are you talking about?

As noted (lines 160-162), we are addressing ‘status-related’ differences in pup aggression for the first time in meerkats. This may be surprising, given the decades of research on meerkats, but it highlights the benefits of detailed focal observation over routine scan sampling, and illustrates one of the unique contributions of our intensive study. This discovery means that we can only speculate on function. Conservatively, we note that high aggression is “coincident with peak competition for attention from and feeding by helpers” (lines 167-170), suggesting that it has the same role in pups as in adults (i.e., feeding competition). We will explore infant development in more detail in the future (see Response 2).

On original line 234, we were talking about the opposite patterns that derive from blocking vs. introducing androgens. We have modified this sentence to expand on the consistency of effects in both sexes and to improve clarity. Notably, we gave the specific example of effects on aggression to better illustrate the ‘reversed’ patterns (lines 280-284).

Line 251: here and above most citations are out of place, but one that demonstrates the effect of the mother best would probably be East et al. (2009), demonstrating with natural cross-fostering that it is basically maternal rank that is important for status

acquisition of cubs, i.e. cubs of low-ranking females cross-fostered by high-ranking females obtain high social status and vice versa. This effect seems to be much larger than the tiny differences in androgen-related agonistic acts of cubs reported by Dloniak et al. (2006).

See Response 3 on citation errors.

See Response 7 on the contribution of both socialization and hormones to infant development, as well as on maternal role. As previously stated, we absolutely agree that socialization is key to infant development, but this study is not focused on maternal socialization effects (nor on matrilineal societies). Again, meerkat mothers play a small role in the socialization of their infants, as other group members rear their offspring. We are specifically interested in the endocrine contribution to behavior. Our selection of a cooperatively breeding species with communal rearing is another of the unique benefits of our study, as it more specifically *isolates* the hormonal effects from maternal socialization effects.

Line 286ff: is there actually evidence for this in meerkats?

We don't understand this question. We are presenting such evidence here.

Line 291ff: Not sure the mechanism described here and referred to Ref 30 is still accurate, i.e. I recommend to check the work by Ned Place, Steve Glickman and coworkers who suggest large genetic effects, i.e. (Glickman et al., 2006; Place et al., 2002).

10. Reviewer 1 has misattributed credit, while ultimately suggesting that I check my own work, notably the original and only experiment on flutamide in pregnant spotted hyenas that led to discovery of both hormonal *and* genetic mechanisms of sexual differentiation:

Drea, C.M., Weldele, M.L., Forger, N.G., Coscia, E.M., Frank, L.G., Licht, P., & Glickman, S.E. (1998). Androgens and masculinization of genitalia in the spotted hyaena (*Crocuta crocuta*). 2. Effects of prenatal anti-androgens. *Journal of Reproduction & Fertility*, 113: 117-127.

Because that study (Ref 42) remains the only flutamide manipulation in hyenas, it features prominently in later reviews (e.g., Glickman et al., 2006), additional analyses (Place et al., 2002, that followed up on the adult animals resulting from this original manipulation), and indeed most Animal Behavior and Endocrinology text books. See bolding below to show Drea's coauthorship on both of the recommended citations.

As first described by Yalcinkaya and colleagues (Ref 32), the hormonal mechanism in hyenas involves a placental route to female masculinization (whereby maternal ovarian A4 is converted to T in the placenta, such that late-gestation T in females exceeds male T (see Response 8) and influences the structural development of the daughters' ovaries; lines 346-350). It remains accurate. Notably, this route has since been discovered in humans (i.e., accounting for a portion of masculinized daughters) and thus Yalcinkaya continues to be cited in the medical literature. Moreover, Drea et al. 1998 experimentally confirmed the endocrine component, while also adding the new genetic component. The latter did not replace the former; instead, the two mechanisms work in conjunction (genetic for the basic phallic structure; endocrine for the sex differences).

References

Dloniak SM, French JA, Holekamp KE, 2006. Rank-related maternal effects of androgens on behaviour in wild spotted hyaenas. *Nature* 440:1190-1193. Already cited.

Glickman SE, Cunha GR, **Drea CM**, Conley AJ, Place NJ, 2006. Mammalian sexual differentiation: lessons from the spotted hyena. *Trends in Endocrinology & Metabolism* 17:349-356.

Not necessary to cite, because we cited the original study (see above).

Place NJ, Holekamp KE, Sisk CL, Weldele ML, Coscia EM, **Drea CM**, Glickman SE, 2002. Effects of prenatal treatment with antiandrogens on luteinising hormone secretion and sex steroid concentrations in adult spotted hyenas, *Crocuta crocuta*. *Biol Reprod* 67:1405-1413.

Not necessary to cite (see above). Notably, the patterns of LH in response to a GnRH challenge, in adult hyenas exposed prenatally to antiandrogens (see Drea et al. 1998), does not detract from flutamide blocking androgen receptors in both sexes and producing consistent *organizational* effects. See Response 2 on scope.

East ML, Honer OP, Wachter B, Wilhelm K, Burke T, Hofer H, 2009. Maternal effects on offspring social status in spotted hyenas. *Behav Ecol* 20:478-483. doi: 10.1093/beheco/arp020.

Goymann W, East ML, Hofer H, 2001. Androgens and the role of female "hyperaggressiveness" in spotted hyenas (*Crocuta crocuta*). *Horm Behav* 39:83-92.

Pribbenow S, Shrivastav TG, Dehnhard M, 2017. Measuring fecal testosterone metabolites in spotted hyenas: Choosing the wrong assay may lead to erroneous results. *J Immunoassay Immunochem* 38:308-321. doi: 10.1080/15321819.2016.1260584.

The prior three studies are also not necessary to cite (see Response 2). Our meerkat study does not require greater review of spotted hyena behavior. Likewise, our study relies principally on serum assays, so does not require detailed methodological consideration of fecal assays.

Reviewer #2 (Remarks to the Author):

The goal of this manuscript is to test an hypothesis suggesting that androgens mediate agonistic interactions among female meerkats as a form of intrasexual competition. The authors combine data on naturally-occurring behavior and hormone concentrations with data obtained from field experiments where several dominant females were treated with anti-androgens (AA) during late pregnancy and early during the postpartum period to assess both activational and organizational effects of androgens on behaviors that function importantly in female meerkat breeding competition, and thus effectively mediate reproductive monopoly in this species. The work is beautifully grounded in the framework developed by Phoenix et al (1959). The authors managed to obtain surprisingly (and delightfully) large sample sizes with respect to their behavioral observations and their sampling of hormone concentrations in both serum and fecal

matter. The logic linking their hypothesis with expected outcomes seems very sound, the quality of the data is very high, and the statistical analyses all seem sound. I also found the metadata tables to be very helpful, so I was glad the authors decided to include those in the main manuscript. The writing is generally clear and succinct, and the author's interpretations of the data seem very reasonable with the exception of the "heritable" component discussed below. The data presented in the Supplementary Materials help rule out various alternative interpretations of the main findings to those on which the authors focus. For instance, although to minimize the impact of this experiment on the study population, dominant control females were not implanted with sham pellets, I found the other types of controls applied (implanting males with both experimental and control pellets, comparing treated females to themselves when untreated, controls applied in the statistical analysis of the data, etc.) to be very convincing. Furthermore, although it is conceivable that the trivial surgical procedure itself uniquely altered the behavior of treated females, this cannot explain the observed changes in the behavior of group-mates documented in the tables in the Supplementary Materials, particularly the behaviors not involving direct interactions with treated females.

We thank Dr. Holekamp for these supportive comments and for recognizing the multiple controls that support our interpretation.

I believe this is an important paper, particularly for its emphasis on a broader role than has been previously recognized for important functions of androgens in mediating reproductive competition among female vertebrates. Whereas the first author has explored the same issues in what many refer to as "sex-role reversed" species such as lemurs and spotted hyenas, the current demonstration of their importance in meerkats exploits a much more tractable study system than would be possible with those other, larger species. Despite some variability in the duration of AA treatments, the tractability of this study system, and the strong and consistent support in their data for their hypothesis, allow the authors to make this paper a real tour de force.

Again, we thank Dr. Holekamp for recognizing the importance of this work.

My only substantive concern is that, although the authors claim that the androgenic mechanism they explore here is heritable, and although it might indeed truly be heritable, the authors have not actually shown that here. Instead, they have shown only the existence of intergenerational transmission of 'masculinized' phenotypes in meerkats. Therefore, the authors either need to remove the word "heritable" from the title and elsewhere in the paper or to actually assess the heritability of this trait using the animal model, with methods described by Kruuk (2004), and as employed to assess heritability of aggressiveness in spotted hyenas by Yoshida et al (2016). This should be fairly straightforward, as the Kalahari Meerkat Project should certainly have all the necessary pedigree data available for these calculations. Using the animal model as Kruuk (2004) suggests would also allow the authors to determine what percent of the overall variation in androgen-mediated aggressiveness is due to heritability, to maternal effects, and what percent remains unexplained. Both are mentioned in this manuscript, but the reader is left wondering about this. If the authors cannot use the animal model

here for some reason, I believe they need to reduce their claim about this trait being heritable to the status of an untested hypothesis.

11. Whereas inherited traits are directly passed down from parents to offspring, heritable traits are not necessarily genetic. Moreover, we're talking about masculinization of *all* female meerkats, just to varying degrees. We therefore viewed our usage of the term as appropriate, but we have changed the term 'heritable' in the title to 'intergenerational' (line 1) to accommodate this concern and avoid any confusion. Elsewhere in the text, we have retained mention of heritability, but have qualified it in the same way that others, including this Reviewer, have done under similar circumstances (i.e., without conducting heritability studies). For example, Dloniak et al. 2006 state that "Maternal effects mediated by prenatal hormone exposure are *potentially important for nongenetic inheritance* of phenotypic traits related to social rank (Schwabl, 1993, PNAS), and thus for shaping individual variation in behaviour and social structure". Beyond similar qualification, we also specified an epigenetic route (lines 95, 122).

Regarding a separate heritability analysis, we appreciate the suggestion, but see Response 2. It would not be straightforward at all, even if the trait were genetically inherited, because unlike spotted hyenas, meerkats are not female philopatric. We have moved relevant information about meerkat maturation and dispersal from the methods to the introduction to make this difference more apparent (lines 66-67). Because dispersing female meerkats are routinely lost to the study population, it would be impossible to accurately assess how many females that ultimately start their own clans and reproduce (successfully) were born to dominant versus subordinate individuals. With increasing use of new telemetry tools to track dispersing animals, that question might be broached in the future.

Misc. smaller issues:

L 75: "would contribute" should be toned down a bit to say "might contribute".

We restructured and toned down the sentence (lines 81-82; adding Ref 28).

L 135-136: "...received more prosociality..." is very awkward. I'd be inclined to replace that with something more like "... received more prosocial overtures from clan-mates..."

Agreed; however, we needed a succinct axis label for figures on prosociality and then applied it throughout for consistency. We've compromised by leaving the axis labels as they were, but elaborating the text accordingly in the manuscript (lines 152, 154, 188) and figure legends (885, 900-901, 914). We thank Reviewer 2 for the suggestion.

Legend to Fig 4: Lines 793-795 don't make sense to me. Why the parenthetical reference to Fig 4 here?

We have amended this passage (lines 916-918).

In the Results section, the use of "dominants versus subordinates" generates some awkward sentences. Why not say (eg., L 149-150) more straightforwardly that "dominant mothers initiated more aggression than did subordinate mothers." Or in line

154 "...from dominant than subordinate mothers..." (rather than dominant versus subordinate mothers).

Done (lines 183-189).

L 178: "Treated matriarchs evicted 2/3 fewer subordinate females than did controls": The fraction here should be replaced by a percentage to make this sentence easier to follow.

Done. We have also added more detail (lines 189-191).

L 234: Reference 36 is cited here yet this paper contains not a single word about "androgen-supplemented females of other species," which is suggested by this citation. This paper discusses rank acquisition in primates and spotted hyenas, but there was no androgen supplementation mentioned in either taxon.

See Response 3. Again, we apologize for the confusion.

The discussion of the importance of timing of androgen exposure (lines 258-276) should certainly cite some of the critical work by Kim Wallen on this in macaques.

I wouldn't want to appear as slighting my PhD advisor, so have added Wallen's 2005 review (Ref 35). To clarify, we originally cited only Goy and colleagues (1988) for their landmark study on this issue, which was particularly focused on effects and timing of androgen exposure in *female* rhesus monkeys. Wallen, as Goy's PhD student, continued this line of work, but focused on androgen exposure in *male* rhesus monkeys (for which the timing issue is identical). Given the journal limits on citation numbers, the more presently germane female focus of the earlier work, and the otherwise duplication on macaques (which do not represent species differences in critical windows), we did not originally think it necessary to cite Wallen.

In Methods, what is the difference between project personnel and non-project personnel, and why is it necessary to make this distinction?

The difference is that project personnel included a team of researchers specific to this study, whereas non-project or KMP personnel included the managerial & Reserve staff overseeing the rotating volunteers who maintain the long-term, life-history records (i.e., different PIs, institutions, funding, methods, research topics, responsibilities, etc.). The distinction was made to accommodate an earlier reviewer, the relevant points being that KMP personnel attribute dominance status independent of project personnel and using specific criteria different to ours (so there was no circularity, lines 363-365) and *all* personnel monitor animal health (so there was no bias in reporting). We have simplified the latter by stating that our on-site vet was involved in health monitoring (line 393).

References cited here

Kruuk, L.E.B. (2004). Estimating genetic parameters in natural populations using the "animal model". *Philos. Trans. Roy. Soc. Lond. B: Biol. Sci.* 359: 873-890.

See Responses 2 and 11.

C. H. Phoenix et al. (1959) Organizing action of prenatally administered testosterone

propionate on the tissues mediating mating behavior in the female guinea pig. *Endocrinology* 65: 369.

Already cited (Ref 27).

Yoshida, K.C.S., Van Meter, P.E. & Holekamp, K.E. (2016) Variation among free-living spotted hyenas in three personality traits. *Behaviour*. 153: 1665–1722.

See Responses 2 and 11.

Kay E. Holekamp

Reviewer #3 (Remarks to the Author):

I enjoyed reading the manuscript by Drea et al and applaud the attempt to understand both the organizational and activational effects of androgens in female meerkats. The paper is well written and thorough. I am not aware of similar investigations in wildlife. The long-term study system that is composed of many clans is ideal to test the interaction between androgens and dominance in females across generations.

We are happy to hear it and thank Reviewer 3 for these positive comments.

The only concern that I have about the study is the lack of validation in females. I appreciate the difficulty involved in validating the procedures on wildlife in the field, especially when part of a long-term research. However, these validations can and should be carried out on captive meerkats.

12. We appreciate the Reviewer's concern about validation; however, we wish to emphasize again that the original submission contained several validations, and we've added additional validations herein (see Response 1).

Unfortunately, it is not possible to conduct validations on captive meerkats. As noted by Reviewer 2, not all animals are equally tractable for a given study, with differences in tractability also differing between captive and wild populations of the same species. The reasons for these differences may not always be obvious. The study at hand is one that is particularly challenging, not only because of the select group of relevant species, but because of the (a) required *a priori* knowledge about female gestational phase, (b) need to minimize anesthesia (particularly in late gestation), (c) required constant flutamide dosing, and (d) need for continuous access to subjects, among other things. Whereas spotted hyenas could be hand fed their dosage twice daily and ultrasounded to facilitate hormonal manipulation in captivity (Drea et al 1998), their size, potential danger to humans, and fission-fusion social system preclude manipulation in the wild (allowing only correlational studies; Dloniak et al. 2006). Lemurs likewise cannot be manipulated in the wild (owing to dosing, access, arboreality, difficult terrain, etc.), and although they logistically could be manipulated in captivity, their endangered status precludes it (again allowing only correlational studies; Grebe et al. 2019).

It is the reverse situation for meerkats: Small enough for implants, they can *only* be studied in the wild (and currently only at the KMP), in this case because captivity in meerkats fundamentally changes the behavior of interest. Notably, socially housed

captive animals (typically in zoos) must be maintained on *ad lib* food to control aggression (i.e., food competition, evictions, etc.), eliminating all potential for relevant comparisons. Moreover, owing to ad lib feeding, captive meerkats are obese, introducing health concerns and additional confounds. Female contraception helps limit animal numbers, but becomes another limiting factor. Because no zoo holds multiple clans, there is only 1 matriarch per zoo, and using multiple zoos would add new confounds (and expense). More to the point, zoos typically decline to facilitate studies perceived as having 'invasive' procedures. Establishing a university colony introduces new hurdles (e.g., obtaining exotic carnivorans, exorbitant costs, disposition of animals, etc.) and would not eliminate any of the major problems just mentioned. In sum, validation in captive meerkats is not a viable option. See Response 1 for the alternative routes taken.

Other comments:

In order to appreciate the effects of androgens in pregnancy and on the offspring, more details describing the system are needed in the Introduction. For example, how many offspring are there on average in a litter? Do dominant females have more female-biased offspring sex ratios?

13. We have added a sentence to address litter size, as requested, as well as some other details on reproductive success (line 65) and have moved information on maturation and philopatry/dispersal from the methods to the introduction (lines 66-67; see also Response 7). Litters are probably not sex biased (MacLeod & Clutton-Brock 2013 *Anim Behav*), but it is impossible to know, given that meerkats are born underground and remain there for a couple weeks, where they may die/be eaten before being sexed (see Dimac-Stohl et al. 2018 and Fig S1). Efforts to ultrasound pregnancies have been undertaken at the KMP, but in litter-bearing species, it is challenging to be certain of litter size and sex ratios. At emergence, the litters, overall, are not sex biased (we did not mention patterns that are either not present or cannot be properly ascertained).

Maybe I missed it, but why was the experimental implantation at 3 weeks? How was this time period chosen? Why not 2 or 4 weeks? How long is meerkat pregnancy? Were implants removed following 3 weeks?

Most of this information was indeed already presented and also illustrated in Fig. S3, but we will address those points here for ease of reference: Three weeks is when pregnancy becomes detectable (line 371), not when treatment began. Our treatment window was the 3rd trimester (ideally the later part of this trimester extending into early postpartum, which is what we achieved; Fig. S3), because this is the time frame during which the neurological substrates underlying behavior are sexually differentiated (lines 82-91, 98-100, 180-181, 309-328). Meerkat gestation lasts 70 days (lines 373-374, 492-493). We selected implants as our method of choice because the pellets, by design, dissolve over their active 3-wk period, releasing a steady dose of flutamide (in our case, ~15 mg/kg/day; lines 465, 492). We have added information about dissolvability for clarification (lines 458-459, 501-502). The active period coincided perfectly with the duration of the third trimester in meerkats (line 491-493), allowing us to require only one

anesthesia during pregnancy. Moreover, as there is nothing to remove after 3 weeks, no anesthesia is required post-partum either.

In a related matter, since there is variability between individuals in the timing that they received the implant (LP or PP), how was the timing of the implant insertion accounted for in the analysis of offspring behaviour?

Time of implant was not directly accounted for; however, litter and maternal ID were accounted for, which would indirectly help control for differences in treatment timing between litters (as well as other litter specific effects, such as e.g., sex ratio, environmental conditions, mother's condition, etc.).

Was year included in the statistic models?

Yes, but as the highly correlated variable of rainfall, which is a more sensitive measure of temporal quality (lines 531-532).

Was pup anogenital distance (AGD) measured when pups were handled? This may provide an indication of earlier androgen exposure and an alternative way to assess in utero masculinization.

As originally stated (lines 313-315), there is no masculinization of female genitalia in meerkats; so AGD would not provide an alternative way to assess in utero masculinization. That said, we did take morphological measurements and those will be presented in a subsequent publication (see Response 2 on scope).

Have the authors related serum and fecal androgen levels? Not sure how related they are, and whether the same outcome would be with serum T, A4. Have the authors measured serum androgens as a result of the implants?

Our normative A4 and T data are from serum (i.e., these assays are specific to the type of steroid), whereas fecal assays measure all androgen metabolites (i.e., the resulting value represents a combined total). To the extent possible, yes, we have 'related' normative serum & fecal data in Fig 1a-c, to show that the status-related and gestational patterns are consistent across all three measures. Our methods and data on serum and fecal assays are provided in Table 1 and in lines 375-378, 395-402, 519-555, and include an explanation for why we did not/could not obtain blood post implant (i.e., 1 knockdown of late-gestation pregnant females in total). See also Response 1. Our fecal androgen concentrations in females are also now compared to those in males (see Response 1). We have also previously published these validations and serum vs fecal comparisons: please see Davies et al., 2016 and DelBarco-Trillo et al. 2016.

The implanting procedure described is invasive and includes pain and anesthesia. Including a control group with empty implants is required in experiments and rules out the possibility that the behavioural effects following insertion are due to the inhibitor rather than pain. I think that such assays also require validation in the same sex as later experimented on, as females are different than males, especially while pregnant.

The procedure is minimally invasive and minimally painful (see Response 1). Blood sampling in wild meerkats requires anesthesia, so *all* of the animals (regardless of treatment) were captured and received anesthesia (see Response 5). A control is required, but need not be an empty implant (see Response 1). Under our circumstances, the lack of a sham in females did not 'rule out the possibility that the behavioral effects owe to the treatment'. Females and males respond equivalently to flutamide (see Response 1).

I found the Discussion short on evolutionary implications. In order to influence thinking in the field, I suggest expanding the Discussion to include additional aspects.

We don't understand this concern. The entire study is about sexual selection in females through understanding the mechanisms underlying female intrasexual aggression, leading to the evolution of female reproductive skew (aka cooperative breeding). As part of the final conversion to *Nature Communication* format, we have added a paragraph (lines 118-123) that should also address this concern. We have tempered our use of the word heritable (see Response 11), but the fact that female masculinization affects all female meerkats means we are addressing the evolution of an unusual phenomenon, namely androgen-mediated female aggression.

I suggest moving Tables 1-3 to the SI.

We thank Reviewer 3 for the suggestion, but like Reviewer 2, we would prefer to keep these tables in the main text, where they will most benefit readers. Our sample sizes are both relevant and noteworthy and should not be relegated to supplementary material.

Figure 2: what data is the model based on? Adding the actual data on the figure may be helpful.

Done (see Response 9 and Supplementary fig 2).

Table S1: Please add measured concentrations of androgens. It would be interesting to see.

Those data are presented in Figure 1, in the main text.

REVIEWERS' COMMENTS

Reviewer #1 (Remarks to the Author):

The ms has greatly improved since the last version. The only concern I have is, as noted earlier, the citation of a paper on fecal androgen measurements in spotted hyenas (ref. 14). As I explained in my previous review, the assay that has been used by this publication cannot distinguish androgen metabolites from glucocorticoid metabolites (see Pribbenow S, Shrivastav TG, Dehnhard M, 2017. Measuring fecal testosterone metabolites in spotted hyenas: Choosing the wrong assay may lead to erroneous results. *J Immunoassay Immunochem* 38:308-321). Therefore, ref. 14 cannot be taken as evidence for high androgen concentrations in female hyenas. Since the authors do not want to cite Goymann et al. 2002 they should at least stick to the citation of Licht et al. (ref 13), which has measured androgen concentrations in plasma and not feces.

Reviewer #2 (Remarks to the Author):

In my opinion, the authors have done a very good job responding to reviewer comments. In my own case they were preaching to the choir, as I was already convinced before they added the three new sets of controls, all of which support their original interpretations. As the authors emphasize in their response document, effects of androgens and maternal socialization on aggression are not mutually exclusive hypotheses; indeed both often occur in the same animals. I agree that the authors' selection of meerkats with their communal rearing is another of the unique benefits of their study, as it more specifically isolates the hormonal effects from maternal socialization effects than would be possible in species where mothers are the sole caregivers.

To me the strongest evidence that antiandrogens were truly effecting the observed changes in aggressive behavior were the unambiguous organizational effects on offspring behavior. The authors have no access to captive meerkats, so that is not an option for them. In fact, I know of not one facility (e.g., a zoo) that houses captive meerkats that would ever permit any sort of manipulation like those carried out here, even though the "surgeries" here were trivial. In my opinion, the authors should not be penalized for something (limitations on sham implanting dominant females) over which they had absolutely no control, particularly when they had so many other types of controls in place. I remain convinced of the importance of this study, and continue to view it as a tour de force, albeit a slightly longer-winded one than the original submission. As two of three reviewers were worried about the lack of sham implants in dominant females, the additional text and controls were clearly needed despite the fact that these additions increased length of the manuscript. Testing the implants on males makes perfect sense to me, because, as the authors point out in their responses, the mechanism of flutamide action is identical in males and females. If the operated meerkats were handicapped by these minor surgeries, we should see differences that reflect this handicap, but we do not. The authors argue appropriately that the scope of their paper should be restricted to their focal topic. They have also nicely clarified their intended meanings wherever asked to do so. I hope my fellow reviewers are reassured by new material in lines 175-223, supplementary figures 2, 4, 5, and 6, and supplementary table 6.

Miscellaneous small things

L 99-100: It should be clearer that the authors refer to a different set of species rather than that some females in "many mammalian species" are "unconventionally aggressive."

L 176: Should this not be "behavioral and endocrine patterns?"

Whereas I recognize that Drea would not slight her PhD advisor and that Wallen's work focused on males, in my own opinion, Wallen's work went well beyond that of his own PhD advisor (Goy), so I'm glad you decided to add this citation.

L 618-619: I hadn't realized that a single female might be evicted multiple times!

Kay E. Holekamp

REVIEWERS' COMMENTS

Reviewer #1 (Remarks to the Author):

The ms has greatly improved since the last version. The only concern I have is, as noted earlier, the citation of a paper on fecal androgen measurements in spotted hyenas (ref. 14). As I explained in my previous review, the assay that has been used by this publication cannot distinguish androgen metabolites from glucocorticoid metabolites (see Pribbenow S, Shrivastav TG, Dehnhard M, 2017. Measuring fecal testosterone metabolites in spotted hyenas: Choosing the wrong assay may lead to erroneous results. *J Immunoassay Immunochem* 38:308-321). Therefore, ref. 14 cannot be taken as evidence for high androgen concentrations in female hyenas. Since the authors do not want to cite Goymann et al. 2002 they should at least stick to the citation of Licht et al. (ref 13), which has measured androgen concentrations in plasma and not feces.

We thank Reviewer 1 for these comments and understand the concern expressed about the fecal assay protocols of other studies, even if they may have produced comparable results to serum/plasma-based studies. As noted by this reviewer, the citation of Licht et al. amply makes our point about androgen concentrations in female hyenas. We would prefer not to elaborate on a topic peripheral to the main focus and also cannot increase the number of citations, so we have removed our citation of the reference in question insofar as it regards spotted hyena fecal androgen concentrations. We retain this citation for all other instances. It has consequently moved from reference 14 to 36 (and all intervening reference numbers have been updated).

Reviewer #2 (Remarks to the Author):

In my opinion, the authors have done a very good job responding to reviewer comments. In my own case they were preaching to the choir, as I was already convinced before they added the three new sets of controls, all of which support their original interpretations. As the authors emphasize in their response document, effects of androgens and maternal socialization on aggression are not mutually exclusive hypotheses; indeed both often occur in the same animals. I agree that the authors' selection of meerkats with their communal rearing is another of the unique benefits of their study, as it more specifically isolates the hormonal effects from maternal socialization effects than would be possible in species where mothers are the sole caregivers.

To me the strongest evidence that antiandrogens were truly effecting the observed changes in aggressive behavior were the unambiguous organizational effects on offspring behavior. The authors have no access to captive meerkats, so that is not an option for them. In fact, I know of not one facility (e.g., a zoo) that houses captive meerkats that would ever permit any sort of manipulation like those carried out here, even though the "surgeries" here were trivial. In my opinion, the authors should not be penalized for something (limitations on sham implanting dominant females) over which they had absolutely no control, particularly when they had so many other types of controls in place. I remain convinced of the importance of this study, and continue to view it as a tour de force, albeit a slightly longer-winded one than the original submission. As two of three reviewers were worried about the lack of sham implants in dominant females, the additional text and controls were clearly needed despite the fact that these additions increased length of the manuscript. Testing the implants on males makes perfect sense to me, because, as the authors point out in their responses, the mechanism of flutamide action is identical in males and females. If the operated meerkats were handicapped by these minor surgeries, we should see differences that reflect this handicap, but we do not. The authors argue appropriately that the scope of their paper should be restricted to their focal

topic. They have also nicely clarified their intended meanings wherever asked to do so. I hope my fellow reviewers are reassured by new material in lines 175-223, supplementary figures 2, 4, 5, and 6, and supplementary table 6.

We would again like to thank Dr. Holekamp for sharing her supportive comments.

We also agree that the requested additions have made the manuscript 'slightly longer-winded' than the original version. We couldn't reduce content, but have incorporated small edits throughout the introduction, results, and discussion to reduce wordiness wherever possible, while also improving clarity. These edits resulted in a roughly 200-word reduction. Additionally, we handled requests from the Editorial Staff regarding additional methodological details on previously published protocols by creating two sections of Supplementary Methods, which modestly helped streamline the methods.

Miscellaneous small things

L 99-100: It should be clearer that the authors refer to a different set of species rather than that some females in "many mammalian species" are "unconventionally aggressive."

Done. We have specified 'those species in which females are unconventionally aggressive also show a signature late-term increase.'

L 176: Should this not be "behavioral and endocrine patterns?"

Done. We have inserted the 'and.'

Whereas I recognize that Drea would not slight her PhD advisor and that Wallen's work focused on males, in my own opinion, Wallen's work went well beyond that of his own PhD advisor (Goy), so I'm glad you decided to add this citation.

Agreed.

L 618-619: I hadn't realized that a single female might be evicted multiple times!

Indeed. While evicted, many females remain in the vicinity of their natal clan and may repeatedly attempt reentry. Although pregnant subordinates present the greatest reproductive threat to the matriarch, they are needed to support the clan, so once they've lost their litter (the typical fate for obligate cooperators), they often return. And they become pregnant more often than is realized, or at least more often than they produce successful litters (mating success is a poor predictor of reproductive success), so the process is repeated.

Kay E. Holekamp